# Evaluating Robustness of Monocular Depth Estimation with Procedural Scene Perturbations

**Jack Nugent, Siyang Wu, Zeyu Ma, Beining Han, Meenal Parakh,**
**Abhishek Joshi, Lingjie Mei, Alexander Raistrick, Xinyuan Li, Jia Deng**
Princeton University
{jacknugent, sw2776, zeyum, bh7032, meenalp,
aj9792, lm5483, araistrick, xl2855, jiadeng}@princeton.edu

## Abstract

Recent years have witnessed substantial progress on monocular depth estimation, particularly as measured by the success of large models on standard benchmarks. However, performance on standard benchmarks does not offer a complete assessment, because most evaluate accuracy but not robustness. In this work, we introduce PDE (Procedural Depth Evaluation), a new benchmark which enables systematic evaluation of robustness to changes in 3D scene content. PDE uses procedural generation to create 3D scenes that test robustness to various controlled perturbations, including object, camera, material and lighting changes. Our analysis yields interesting findings on what perturbations are challenging for state-of-the-art depth models, which we hope will inform further research. Code and data are available at `https://github.com/princeton-vl/proc-depth-eval`.

## 1 Introduction

**The need for robustness evaluation.** Monocular depth estimation refers to the task of producing a per-pixel depth map from a single image. It is a key task in 3D vision. It is especially important when only a single image is available, or multiple images are available but parts of the scene are motionless relative to the camera.

In recent years, monocular depth estimation has seen significant advancements. Most notable is the success of large models geared toward *general* monocular depth estimation. These models are designed and trained with the goal of performing well on *arbitrary* scenes. Accuracy on standard benchmarks such as KITTI[10], NYU-D[33], ETH3D[32] and DIODE[37] have seen large improvements.

However, standard benchmarks do not offer a complete assessment of performance, as most only evaluate accuracy, not robustness. They do not tell us whether the predictions would remain reliable under perturbations of the scene content. Would the estimated depth change significantly if the light comes from below instead of above? What if the material is more specular? What if the object has a slightly different shape? What if the camera moves slightly away? Existing depth robustness evaluations [18] do not address these changes, instead focusing only on 2D image corruptions. Assessing robustness to camera, object, lighting and material changes is important because they are prevalent in downstream applications such as robotics and view-synthesis. A model that is insufficiently robust is prone to unpredictable performance degradation, is vulnerable to adversarial attacks, and cannot be deployed in mission-critical applications.

In other domains such as natural language reasoning and visual question answering [39, 38, 47, 12], it has been revealed through targeted evaluation that many leading foundation models can struggle under small perturbations of the input questions. Such evaluations have provided valuable insights

39th Conference on Neural Information Processing Systems (NeurIPS 2025).

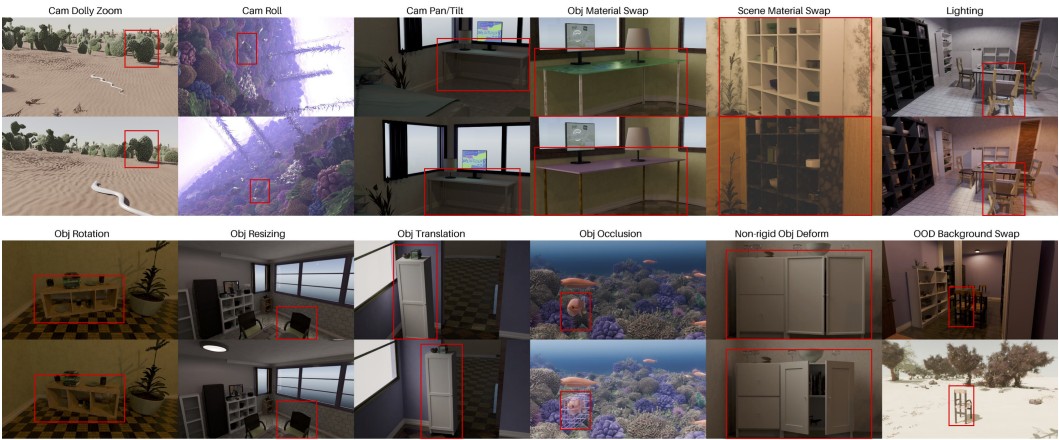

Figure 1: **Perturbation Types** We use procedural generation to construct controlled perturbations which vary one aspect of a scene while keeping others constant. We show a single example for each perturbation type, with the object of interest highlighted by red boxes.

for further research. Our work aims to provide similar insights for the development of robust depth estimation models.

**Robustness evaluation with procedural perturbations.** Our work leverages synthetic data to systematically evaluate the robustness of monocular depth estimation models. We use procedural generation to create many diverse scene perturbations. Each perturbation type changes one aspect of the scene while keeping all others constant. The types of perturbations include moving the camera slightly, changing the camera intrinsics, changing lighting, changing the material, deforming the shape through articulation, among many others. A full list of perturbations can be found in Table 1 and examples of such perturbations are shown in Fig. 1.

We evaluate using synthetic indoor and outdoor scenes generated by Infinigen [27, 28], a photorealistic open-source procedural generator. We implement our perturbations as additional procedural generation code on top of this system. Infinigen scenes are computer rendered, and thus have some domain gap, but are suitable for robustness evaluation as humans can still easily perceive their depth. Infinigen scenes are also ideal for our purpose because they have high quality ground truth and are not used as a source of training data by any of the models we evaluate, facilitating precise and unbiased evaluation. In addition, the procedural nature of Infinigen virtually guarantees that the test scenes are novel and thus suitable for assessment of generalization and robustness. Finally, Infinigen objects are fully procedural, enabling non-rigid and semantically meaningful deformation of shapes (e.g. elongating the legs of a chair) that would be otherwise cumbersome to achieve.

We evaluate 11 state-of-the-art monocular depth models: MiDaS[2], DepthAnything[44], DepthAnything V2[45], ZoeDepth[1], UniDepthV2[25], Metric3D V2[14], Marigold[16], DepthPro[3], MoGe[41], GeoWizard[8], and VGGT[40]. We test each model on a set of 3D scenes plus their associated procedural perturbations. For each scene, we observe how the predicted depth changes under the perturbations, in particular, how much the predictions would differ from each other as well as how much the error metrics against ground truth would change. We have released open source code to allow our evaluation to be reproduced and repeated for new depth estimation models.

**Accuracy Stability versus Self-Consistency.** We consider two related but different notions of robustness: accuracy stability and self-consistency. Accuracy stability is the variance of the difference between a model's prediction under perturbation against the ground truth, reflecting the degree to which the accuracy remains stable. Self-consistency is measured as the average (squared) difference between a model's prediction under perturbation and the original prediction, reflecting how well the predictions remain stable as compared to its own original prediction, regardless of ground truth.

Self-consistency is generally stricter than accuracy stability. As long as the perturbations do not change the ground truth geometry, perfect self-consistency implies perfect accuracy stability, but not vice versa: it is possible for the model to make a set of predictions under perturbation that differ from the original prediction (thus not self-consistent) but all have the same numeric accuracy against

ground truth (thus perfectly accuracy stable). On the other hand, accuracy stability is more broadly applicable: self-consistency is ill-defined when perturbations such as non-rigid deformations alter the ground truth geometry, meaning that the depth predictions of an accurate model necessarily differ from each other.

**Main Findings.** Our evaluation yields several interesting findings. We observe that state of the art depth estimation models are relatively robust to perturbations in lighting and 3D object pose, but are less robust to occlusion and material changes. Surprisingly, camera perturbations were also among the most challenging: camera rotation and focal length change are among the most difficult perturbations, much more difficult than perturbations to the 3D object pose. We report results for all listed models and perturbation types, including many qualitative examples, which we hope will inform further development of robust depth estimation models.

## 2 Related Work

**Monocular Depth Estimation.** Recent years have seen a large number of monocular depth estimation models [2, 44, 45, 1, 24, 25, 14, 16, 3, 41] that have achieved impressive results on existing benchmarks. This large selection of models provides rich choices for downstream applications, but their robustness remains unclear because benchmark performance alone does not offer targeted and controlled evaluation of robustness. Our work provides this evaluation.

**Monocular Depth Benchmarks.** Most monocular depth benchmarks begin with collecting a large dataset of real-world image captures[10, 33, 32, 37]. Others, such as [4], are made from human-crafted synthetic data. In either case, standard benchmarks lack paired evaluation scenes with controlled perturbations, and thus cannot evaluate robustness metrics such as accuracy stability and self-consistency.

Closest to our work is RoboDepth[18], which evalutes the robustness of monocular depth models to image-level corruptions such as sensor noise, blur, color and added particles. These are orthogonal to our procedural perturbations of scene content such as object, camera, material and lighting changes. Robustness to these perturbations is important for downstream tasks and to our knowledge has not seen targeted evaluation in any prior work.

**Synthetic Data for 3D Vision.** Synthetic data rendered through computer graphics has seen growing adoption in computer vision and AI research, driven by its ability to be generated in unlimited quantities while providing high-quality 3D ground truth. In 3D vision, synthetic data has been widely used to create diverse environments, including natural landscapes[4, 42, 23], indoor settings[5, 46, 7, 30], and city-scale scenes[6, 43, 17], enabling large-scale training of computer vision models and embodied agents. Notably, many state-of-the-art 3D vision and robotic systems trained purely in simulation have demonstrated remarkable zero-shot performance in real-world scenarios[35, 36, 15, 21]. Our benchmark is built on Infinigen[27, 28], a recent synthetic data generation system that enables procedural generation with fine-grained control.

**Robustness Evaluation.** Since the introduction of adversarial examples[34], researchers have sought to understand and leverage their power in computer vision. In the field of 2D vision, studies have explored why adversarial examples would arise, either from the internal structures of neural networks[11], or from signals in the physical world[19]. In other domains, such as Visual Language Models (VLMs), adversial examples can lead to incorrect object groundings in visual understanding[9], or to jailbreak an aligned LLM, resulting in harmful instructions[26]. Additionally, efforts have been made to compile adversarial examples into benchmarks to assess model robustness in both 2D vision[13] and VLMs[47].

## 3 Method

We construct our PDE (Procedural Depth Evaluation) dataset using scenes generated by both Infinigen Nature[27] and Infinigen Indoors[28]. Each scene is generated from scratch using procedural generation (randomized python code), which allows complete control and customization of every aspect of the scene. We generate scenes containing an object of interest from one of the following categories: *chairs*, *desks*, *cabinets*, *fish*, and *cacti*. We chose these categories to encompass a diverse range of both natural and artificial objects, each with distinct geometric characteristics. We ensure

| Perturbation Type | Properties changed in the Perturbed Scene | | | | | |
|---|---|---|---|---|---|---|
| | Camera Intrinsics | Camera Pose | 2D Object Depth Image Modulo (SE(2)) | Light | Material | Background Depth Image |
| Cam Dolly Zoom | Y | Y | Y | | | Y |
| Cam Roll | | Y | | | | Y |
| Cam Pan/Tilt | | Y | Y (homography) | | | Y |
| Obj Material Swap | | | | | Y | |
| Scene Material Swap | | | | | Y | |
| Lighting | | | | Y | | |
| Obj Rotation | | | Y | | | Y (minor) |
| Obj Resizing | | Y | | | | Y |
| Obj Translation | | | Y | | | Y (minor) |
| Obj Occlusion | | | Y | | | Y (minor) |
| Non-rigid Obj Deform | | | Y | | | Y (minor) |
| OOD Background Swap | | | | | | Y |

Table 1: An overview of our procedural perturbations. We introduce 12 distinct scene perturbations, each causing varying degrees of modification to the scene. Examples of individual perturbations can be found in Fig. 1.

the object of interest is salient in the scene, but it may have different appearance or be occluded when we apply perturbations. In total, the PDE dataset comprises 5 object categories and 38 distinct scenes, with each object category appearing in 8 scenes. In the next section, we will introduce 12 possible procedural perturbations, each with up to 60 different parameter settings. This results in a total of 13684 unique scene variations, offering a diverse and comprehensive evaluation of model robustness.

## 3.1 Procedural Perturbations

**Perturbations Types.** We implemented a diverse set of perturbations, covering many aspects of scene content and image formation, including camera intrinsics, camera pose, material, lighting, and 3D geometry. In each scene, we select an object of interest which we ensure is affected by the perturbation but remains salient and within the camera frustum. This object of interest enables finer-grained object-level assessment of robustness as opposed only on the full image. The rest of scene is considered the background.

- **Camera Dolly Zoom**: We change the focal length of the camera (i.e. zoom in or out) and move the camera forward or backward such that the object remains roughly the same size in 2D.

- **Camera Roll**: We rotate the camera around its optical axis, keeping the camera center fixed. This is equivalent to rotating the image around the principal point.

- **Camera Pan/Tilt:** We rotate the camera around the Y-axis (panning) or the X-axis (tilting), while keeping the camera center fixed. In other words, the camera looks up or down, left or right with pure rotation.

- **Object Material Swap:** We randomly switch the material of the object of interest to a new one, for example, from wood to metal.

- **Scene Material Swap:** We randomly switch the material of every object in the scene.

- **Lighting:** We include several variations for lighting including adding or removing light sources, changing strength and temperature, and changing the type of light (i.e., point and spot lights).

- **Object Rotation:** We rotate the object around the origin of its local reference frame.

- **Object Translation:** We move the object without changing its rotation.

- **Object Resizing:** We scale the object to make it larger or smaller relative to the rest of the scene. At the same time, we move the camera center such that the 2D projection of the object consists of exactly the same pixels in the image.

- **Occlusion:** We add a cage-shaped object around the object of interest. We vary the number of lines and the thickness to produce a random 10-40% of occlusion of the object.

- **Non-rigid Object Deformation:** We deform the object of interest in non-rigid ways. Some are commonly occurring, such as opening the doors of a cabinet. Some are unusual and out of distribution, such as a cabinet with a twisted shape.
- **Out-of-distribution (OOD) Background Swap:** We place the object in an unusual background scene, such as a chair underwater. The object pose is identical relative to the camera.

**Effects of perturbation on ground truth depth.** Different perturbations affect ground truth depth differently. Some perturbations, such as lighting change or material swap, do not change the ground truth. A robust model should then predict the same depth. In such cases, the perturbation can be made drastic to stress test the robustness of models. Other perturbations, such as object rotation or translation, do change the ground truth depth. In such cases, we limit the perturbations to a small neighborhood so that the changes to ground truth depth are not so drastic that the model is effectively estimating depth for an unrelated object.

Note that when a perturbation changes the ground truth depth of the object of interest, it changes the object's *depth image*, consisting of the 2D mask of the object plus the depth values. For example, camera pan/tilt changes both the 2D mask (by a homography) and the depth values within the mask.

For perturbations that change ground truth, we can evaluate accuracy stability (how much the error against ground truth varies). But self-consistency (how much the prediction varies as compared to the unperturbed scene independent of ground truth) is ill-defined because the prediction is supposed to vary under such perturbations. So we only evaluate self-consistency on perturbations that do not change the ground truth, except for the two following special cases: (1) **Camera Roll**: It induces a $SE(2)$ transform on the depth image of the object, i.e. the depth image is changed but only up to rotation and translation. We evaluate self-consistency by accounting for this rotation and translation. (2) **Object Resizing**: Though the object is made larger or smaller compared to its background geometry, the camera center is moved such that 2D projection of the object remains the same, so does its depth image (up to scaling of depth values).

Table 1 summarizes the effects of each perturbation, including whether each perturbation changes the depth image of the object of interest modulo $SE(2)$ transforms. Perturbations that do not change the ground truth depth image (object, background, or scene) are suitable for the evaluation of self-consistency.

## 3.2 Evaluation Metrics

Our evaluation of robustness builds upon standard depth error metrics already in use in existing literature: *AbsRel*, $Log_{10}$[31], and $\delta_1, \delta_2, \delta_3$ scores[20].

Let $\Delta(x_1, x_2)$ be a depth error metric that computes the difference between two depth maps $x_1$ and $x_2$. Given a base scene and $N$ perturbations, let $x_0$ be the depth prediction of a model on the base scene, and let $x_i, i = 1, \ldots, N$ be the depth predictions on the perturbed scenes, and let $\hat{x}_i, i = 0, \ldots, N$ be the corresponding ground truth depth maps. We compute three metrics:

- **Average Error** $\mu$**:** the depth prediction error against the ground truth averaged over the perturbed scenes plus the base scene.
$$\mu = \frac{1}{N+1} \sum_{i=0}^{N} \Delta(x_i, \hat{x}_i)$$

- **Accuracy (in)stability** $\sigma$**:** the sample variance of the depth errors over the perturbations and the base scene.
$$\sigma = \frac{1}{N} \sum_{i=0}^{N} (\Delta(x_i, \hat{x}_i) - \mu)^2$$

- **Self-(in)consistency** $\kappa$: the average squared difference between a prediction under perturbation and the prediction on the base scene.
$$\kappa = \frac{1}{N} \sum_{i=1}^{N} \Delta(x_i, x_0)^2$$

Each of the three above metrics can be evaluated on the object of interest or the full scene. By default, we evaluate on the object of interest unless otherwise noted.

| Perturbation Type | Metric | MiDaS | DAv1 | DAv2 | ZoeD | UniDV2 | Metric3DV2 | Marigold | DepthPro | MoGe | GeoWizard | VGGT | Avg |
|---|---|---|---|---|---|---|---|---|---|---|---|---|---|
| Cam Dolly Zoom | Avg. Err. (↓) | 2.41 | 2.04 | 1.84 | 2.74 | 1.47 | 1.84 | 2.44 | **1.18** | 1.47 | 2.31 | 1.99 | 1.97 |
| | Acc. (In)Stab. (↓) | 0.63 | 0.53 | 0.50 | 0.79 | 0.44 | 0.48 | 0.76 | **0.35** | 0.44 | 0.72 | 0.57 | 0.56 |
| Cam Roll | Avg. Err. (↓) | 2.44 | 2.15 | 1.89 | 2.62 | 1.71 | 1.81 | 2.40 | **1.70** | 1.84 | 2.25 | 2.00 | 2.07 |
| | Acc. (In)Stab. (↓) | 0.47 | 0.44 | 0.47 | 0.41 | **0.32** | 0.37 | 0.44 | 0.50 | 0.34 | 0.41 | 0.33 | 0.41 |
| Cam Pan/Tilt | Avg. Err. (↓) | 2.20 | 1.80 | 1.61 | 2.42 | 1.28 | 1.61 | 2.09 | **1.11** | 1.33 | 2.00 | 1.74 | 1.74 |
| | Acc. (In)Stab. (↓) | 0.48 | 0.31 | 0.32 | 0.45 | **0.25** | 0.32 | 0.41 | 0.30 | 0.27 | 0.37 | 0.30 | 0.34 |
| Obj Material Swap | Avg. Err. (↓) | 2.29 | 1.84 | 1.64 | 2.41 | 1.32 | 1.69 | 2.11 | **1.16** | 1.42 | 2.00 | 1.76 | 1.79 |
| | Acc. (In)Stab. (↓) | 0.40 | 0.32 | 0.24 | 0.40 | 0.24 | **0.23** | 0.32 | 0.25 | 0.34 | 0.33 | 0.34 | 0.31 |
| Scene Material Swap | Avg. Err. (↓) | 2.28 | 1.85 | 1.62 | 2.42 | 1.31 | 1.64 | 2.11 | **1.29** | 1.46 | 2.06 | 1.78 | 1.80 |
| | Acc. (In)Stab. (↓) | 0.45 | 0.41 | **0.31** | 0.44 | 0.35 | 0.32 | 0.38 | 0.45 | 0.43 | 0.38 | 0.42 | 0.39 |
| Lighting | Avg. Err. (↓) | 2.19 | 1.83 | 1.60 | 2.34 | 1.35 | 1.69 | 2.05 | **1.10** | 1.35 | 1.98 | 1.68 | 1.74 |
| | Acc. (In)Stab. (↓) | 0.25 | 0.15 | 0.13 | 0.23 | **0.13** | 0.16 | 0.22 | 0.19 | 0.17 | 0.24 | 0.18 | 0.19 |
| Obj Rotation | Avg. Err. (↓) | 2.18 | 1.81 | 1.65 | 2.35 | 1.36 | 1.66 | 2.10 | **1.16** | 1.40 | 2.03 | 1.78 | 1.77 |
| | Acc. (In)Stab. (↓) | 0.36 | 0.30 | 0.29 | 0.35 | 0.28 | 0.28 | 0.36 | **0.26** | 0.33 | 0.36 | 0.30 | 0.32 |
| Obj Resizing | Avg. Err. (↓) | 2.19 | 1.82 | 1.64 | 2.31 | 1.32 | 1.67 | 2.03 | **1.07** | 1.39 | 2.01 | 1.78 | 1.75 |
| | Acc. (In)Stab. (↓) | 0.20 | 0.16 | **0.13** | 0.24 | 0.15 | 0.17 | 0.24 | 0.13 | 0.19 | 0.26 | 0.19 | 0.19 |
| Obj Translation | Avg. Err. (↓) | 2.24 | 1.86 | 1.64 | 2.36 | 1.35 | 1.68 | 2.11 | **1.14** | 1.39 | 2.01 | 1.78 | 1.78 |
| | Acc. (In)Stab. (↓) | 0.32 | 0.24 | 0.21 | 0.33 | **0.20** | 0.21 | 0.35 | 0.23 | 0.25 | 0.32 | 0.28 | 0.27 |
| Obj Occlusion | Avg. Err. (↓) | 2.34 | 2.03 | 1.92 | 2.42 | 1.64 | 1.87 | 2.44 | **1.54** | 1.57 | 2.26 | 1.97 | 2.00 |
| | Acc. (In)Stab. (↓) | 0.38 | **0.29** | 0.30 | 0.36 | 0.33 | 0.31 | 0.46 | 0.36 | 0.36 | 0.38 | 0.38 | 0.36 |
| Non-Rigid Obj Deformation | Avg. Err. (↓) | 2.13 | 1.79 | 1.63 | 2.34 | 1.32 | 1.65 | 2.05 | **1.18** | 1.38 | 1.97 | 1.76 | 1.75 |
| | Acc. (In)Stab. (↓) | 0.44 | 0.34 | **0.30** | 0.48 | 0.30 | 0.31 | 0.43 | 0.32 | 0.34 | 0.41 | 0.37 | 0.37 |
| Average | Avg. Err. (↓) | 2.26 | 1.89 | 1.70 | 2.43 | 1.40 | 1.71 | 2.18 | **1.24** | 1.45 | 2.08 | 1.82 | 1.83 |
| | Acc. (In)Stab. (↓) | 0.40 | 0.32 | 0.29 | 0.41 | **0.27** | 0.29 | 0.40 | 0.30 | 0.31 | 0.38 | 0.33 | 0.34 |

Table 2: Performance of depth estimation models across different perturbations, measured in the error and stability of the AbsRel metric. Best results are highlighted in bold.

| Perturbation Type | Metric | ZoeD | UniDV2 | Metric3DV2 | DepthPro | MoGe | VGGT | Avg |
|---|---|---|---|---|---|---|---|---|
| Cam Roll | Self Con. (↓) | 3.78 | **1.84** | 2.33 | 2.12 | 2.04 | 3.17 | 2.55 |
| Obj Material Swap | Self Con. (↓) | 3.26 | 1.45 | 1.80 | 1.45 | **1.42** | 2.31 | 1.95 |
| Scene Material Swap | Self Con. (↓) | 3.22 | 1.66 | 1.91 | 1.75 | **1.62** | 2.57 | 2.12 |
| Lighting | Self Con. (↓) | 2.25 | 1.19 | 1.43 | 1.26 | **1.08** | 1.81 | 1.50 |
| Obj Resizing | Self Con. (↓) | 1.79 | **0.86** | 1.16 | 0.93 | 0.87 | 1.69 | 1.22 |
| Average | Self Con. (↓) | 2.86 | **1.40** | 1.73 | 1.50 | 1.40 | 2.31 | 1.87 |

Table 3: Self-consistency of the AbsRel metric across depth estimation models and perturbations. Best results are highlighted in bold.

**Self-consistency and affine-invariant depth:** We do not compute the self-consistency metric for methods that predict affine-invariant depth, i.e. depth ambiguous up to an unknown scale and shift. This is because for self-consistency, we need to compare a prediction under perturbation against the base prediction, independent of ground truth. But, it is unclear how to choose the proper scale and shift for the base prediction without referencing the ground truth, unless the error metric itself is invariant to such choices, which unfortunately is not the case for the standard metrics we evaluate.

### 3.3 Evaluation Methods

We evaluate all models on our dataset consisting of $1280 \times 720$ images. We use the default inference procedure of each model (which may include image resizing) to output a depth map of the same resolution. We follow standard procedures for computing the scale and shift alignment as in Marigold[16] and MiDaS[29]. When evaluating on an object of interest, we compute alignment using only the depth values for the object. Additional details can be found in the section A of the appendix.

## 4 Analysis

Tables 2 and 3 present our main results of average error, accuracy (in)stability, and self-(in)consistency of absolute relative error metric. Results for other metrics closely match the AbsRel metric and are provided in the appendix.

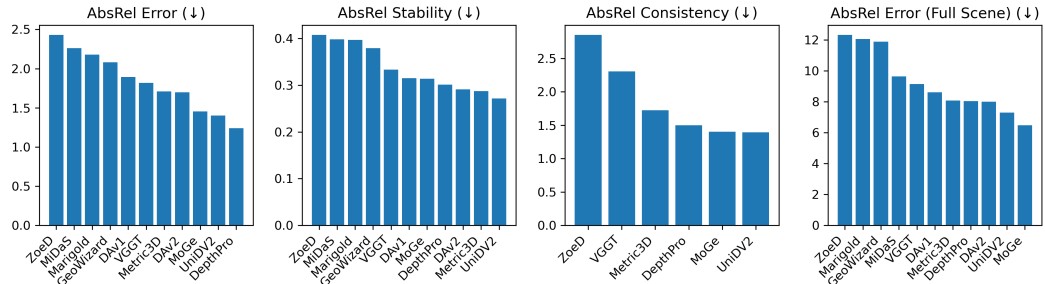

Figure 2: Depth estimation models ranked by Average Error, Accuracy (In)Stability and Self-(In)Consistency on the object of interest, and by AbsRel on the full scene including background.

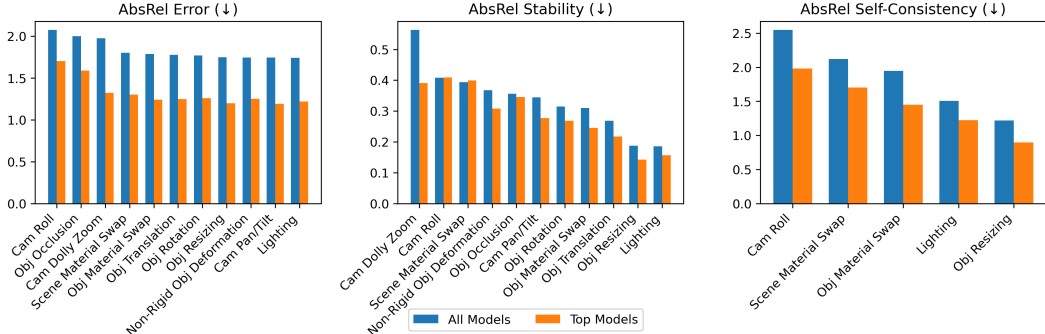

Figure 3: Variations ranked by Average Error, Accuracy (In)Stability, Self-(In)Consistency on the object of interest. *Top models* shows relative perturbation difficulty averaged over DepthPro and UniDepthV2, the two best performing models by AbsRel Error.

## 4.1 Overall Robustness of Models

Figure 2 ranks the models by our main metrics: average error, accuracy (in)stability, and self-inconsistency, computed only on the object of interest and aggregated over all perturbations. We find that DepthPro has the lowest error against the ground truth by a margin of about 13% and achieves the lowest error on every variation type. However, UniDepthV2 and Metric3DV2 display the best stability results. Among scale-invariant models for which self-consistency applies, UniDepthV2 achieves the best score, followed closely by MoGe. Hence, though DepthPro can best predict the detailed depth maps of the objects, it is less robust to perturbations than other MDE models. In the following section we see that no model is able to achieve precise depth predictions while maintaining robustness. This section focuses on evaluation on individual objects; in section B of the appendix we provide more details on models' performance on the entire scene.

## 4.2 Robustness by Perturbation Type

Figure 3 ranks the perturbation types by the difficulty they present to the models. Surprisingly, we see that camera dolly zoom (focal length change) and camera roll (image rotation) are quite difficult. Conversely, the models are very robust to lighting changes, even though there are strong shadows in our lighting perturbation.

**Camera Changes** We observe that camera changes often induce strong distortions in the predicted depth even though the visual information in the scene is unchanged. These often take the form of "warping" from an accurate depth map to something physically implausible. Figure 4 displays two such instances. For camera roll, Depth Pro performed best with an average error of 1.70, but its instability score of 0.50 is considerably worse than UniDepthV2's score of 0.32. Scene "A" follows this trend, with DepthPro providing better estimations than UniDepthV2 for small angles and worse estimations for large angles. The distortions of the fish are severe, as DepthPro predicts that the center of the fish is the closest to the camera when the head of the fish is actually the closest. For

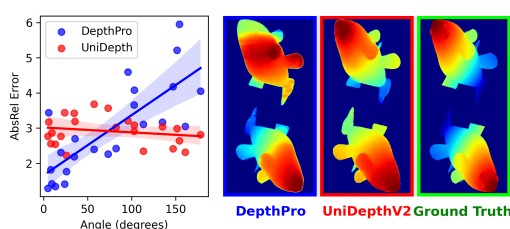
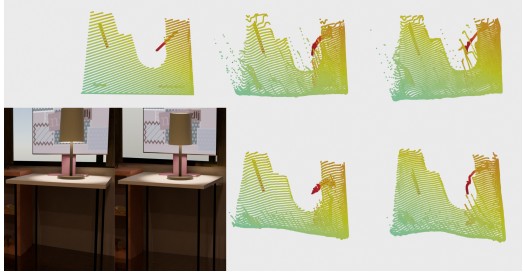

(a) Camera Roll: Predictions from DepthPro and UniDepthV2 on fish scene "A." The depth maps for the fish are cropped from a rotated image (top) and the standard image (bottom)

(b) Dolly Zoom: the top row displays the ground truth point cloud and MoGe predictions on images (1) and (2). The bottom row displays images (1) and (2), and DepthPro predictions on images (1) and (2)

Figure 4: Camera Pose Robustness

the camera dolly zoom variation, Depth Pro achieves the best error and stability scores, but is still far from perfect. In the desk example, we see how a different focal length induces warping in Depth Pro's prediction for the shape of the desk. In this instance MoGe does not display the same fault, though its overall accuracy and stability scores are each about 25% worse than DepthPro for the focal length variation.

**Lighting** Our findings indicate that depth estimation models exhibit strong robustness to lighting perturbations, as lighting had low error and the second-best stability and self-consistency scores among all models.

**Material** We find that models are not able to generalize to novel combinations of materials and shapes, with many models reporting high error and instability for the "scene material swap" and "object material swap" variations. Though these swaps may create implausible combinations (e.g. a floor tiles material applied to a chair), this should not impair depth perception. The DepthAnythingV2 model exhibits strong robustness to these variations, perhaps indicating a success of their large scale training dataset of "pseudo-labeled" real images [45].

**Non-rigid Object Deformation** In contrast to material alterations, the non-rigid object deformation variation creates scenes that differ in semantically meaningful but complex manners. Depth Pro performs the best overall, but we see in figure 5 that the trend differs by object type. Depth Pro has the best error and stability for chairs, which requires predicting the fine structure of curved chair arms or the precise angle of the chair back. However, Depth Pro has worse stability and slightly worse error on desks and cabinets, where the main challenge is predicting the precise location of corners and 90-degree angles.

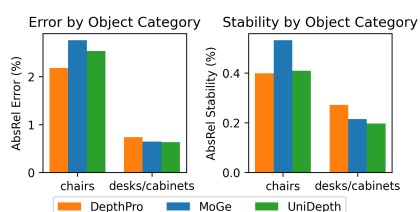

Figure 5: Non-Rigid deformation performance by object category.

**Object Rotation, Translating, Scaling** These models are fairly robust to small Euclidean transformations of the object. Object resizing was nearly the easiest variation by error stability and self-consistency. The (in)stability scores of translation and rotation variations are likewise relatively low.

**Occlusion** Another challenging variation for models is occlusion, which has the second-highest overall error. Moreover, the top models are not able to improve on the error stability of occlusion predictions, indicating that it remains an unsolved problem. Occlusion poses a unique challenge, since it requires understanding the object behind the thin wires. One failure case of DepthAnythingV2 is displayed in figure 6. Although it correctly understands the shape of the chair in the bottom instance with few occluding bars, in the top instance it predicts that the chair seat is nearly vertical. This is physically implausible since the chair seat is connected to the chair's front legs – something humans understand implicitly.

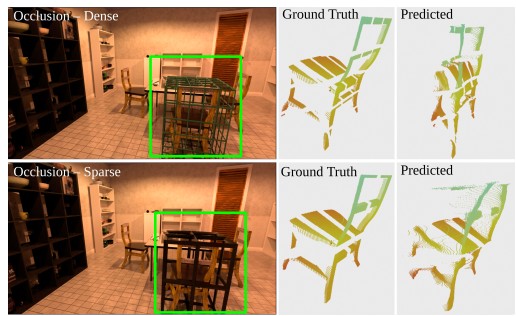

Figure 6: Occlusion Robustness – We visualize point cloud comparisons between DepthAnythingV2 and ground truth depth for a chair occluded by thin structures.

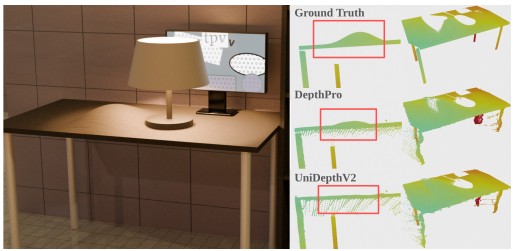

Figure 7: OOD Deformation Robustness – We visualize point clouds from DepthPro and UniDepthV2 for a desk with an unusual added bump. DepthPro attempts to reconstruct it whereas UniDepth predicts a mostly flat surface.

**OOD Deformations** In addition to the procedurally generated variations described above, we manually create objects with intentionally odd geometry that would not appear in the real-world. We show one such example in figure 7. Here, the subtle but human-perceptible change is a bump added to the desk. DepthPro perceives the bump but does not predict precisely the right scale. UniDepthV2, however, predicts the desk to be completely flat, aligning more closely to the expected distribution of real-world objects. This follows a similar trend to the non-rigid object deformations, where DepthPro performs better at recognizing the fine details in the structure of chairs.

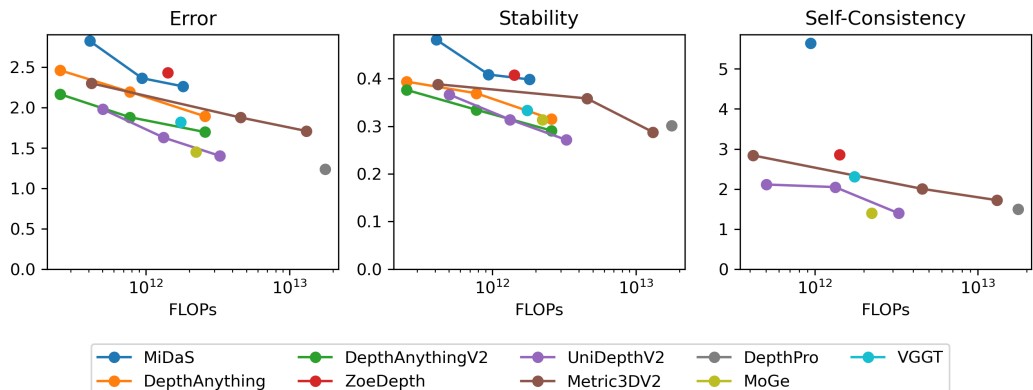

Figure 8: Model performance by FLOPs. Lines indicate a "family" of models (e.g. UniDepthV2-S,M,L). Marigold (477 TFLOPs, 2.18 Error, 0.40 Stability) and GeoWizard (941 TFLOPs, 2.08 Error, 0.38 Stability) are excluded for visualization clarity.

### 4.3 Model Characteristics

Among modern depth estimation models, there is no single architecture or approach that dominates. Though many works including MiDaS, DepthAnything, and DepthAnythingV2 rely on affine-invariant depth as part of their training and output, we find that top-performing models Depth Pro and UniDepth directly predict metric depth. As shown in figure 8, these best performing models are unsurprisingly the largest. For smaller models, however, we find that DepthAnythingV2 variants are quite strong, and their stability exceeds similarly sized variants of UniDepthV2. However, the best model under 1 TFLOP (DepthAnythingV2-Base) still has a 52% higher error than DepthPro, highlighting the need for accurate and efficient depth estimation models.

## 5 Limitations

Our evaluation could be expanded to include additional categories of 3D objects, to improve coverage, or add more categories of scenes such as urban or driving scenes. We focus primarily on changes to

the 3D content of the scene which is rendered with a near-ideal camera; future work could further explore how different forms of camera sensor noise impact performance. Lastly, our evaluation utilizes photorealistic synthetic data for evaluation and our results may not directly translate into the real world. In this sense, our work is best utilized as a method for identifying weaknesses in monocular depth estimation models. Other evaluations would be necessary to certify a model's robustness in the real world. We further discuss this in section C of the appendix.

## 6    Conclusion

We use procedural generation to perform depth robustness evaluation with controlled perturbations in object pose, cameras, materials and lighting. Our findings reveal which models are most robust to these perturbations, and which perturbations are broadly most challenging, both of which inform further research.

## 7    Acknowledgments

This work was partially supported by the National Science Foundation and an Amazon Research Award.

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

# A  Evaluation Details

For each depth prediction, we have three different choices for our evaluation procedure

- Alignment: we either compute a scale factor for the predicted depth or a scale and shift factor.
- Depth for comparison: we compare with the ground truth depth for the error and error instability, or the model's prediction on the base scene for the self-inconsistency score.
- Evaluation mask: we either evaluate on the object of interest or the entire scene excluding the background.

In all cases, the pixels identified by the evaluation mask are the same as the pixels used for calculating the alignment.

**Depth Alignment:** For models which produce a depth estimate as the output, we calculate the alignment parameters using least squares error with the ground truth. We then update the model's prediction $\mathbf{d}$ as $\mathbf{d}^* = a\mathbf{d} + b$ for the scale and shift alignment setting or $\mathbf{d}^* = a\mathbf{d}$ for the scale alignment setting. We calculate the absolute relative error or another metric using $\mathbf{d}^*$. The error and alignment are calculated on pixels in the object mask for the "object" setting or on all pixels not in the background for the "full scene" setting. We also clip $\mathbf{d}^*$ to the dataset range of a minimum depth of 10 centimeters to a maximum depth of 1000 meters.

**Disparity Alignment:** For models which produce a disparity estimate as the output, we calculate the alignment parameters with reference to the ground truth disparity. Then, for a disparity prediction $\boldsymbol{\rho}$ we compute $\mathbf{d}^* = 1/(a\boldsymbol{\rho} + b)$. All other details are identical to depth alignment.

**Self-Consistency Alignment:** When computing self-consistency, the scale of the depth prediction on the base image can vary between models. This does not affect the AbsRel calculation, but does impact other metrics. We normalize the prediction on the base image to have a median depth of 1 meter. Then we calculate the alignment parameters relative to this normalized depth. We do not apply a minimum or maximum threshold for depth, as the depth no longer corresponds to metric distances.

**Object Masks:** We use the object masks provided by Infinigen. However, for pixels on the boundary of an object, it may be unclear whether they belong to the object or the background. This ambiguity may result in improper alignment, since a depth prediction for the background image will differ substantially from a depth prediction for the object. In order to avoid this, we erode the object masks by one pixel.

**Background Masks:** We define the background as anything at infinite distance (e.g. the sky). For underwater scenes, there are some objects at finite distance which are significantly obscured by the water. For these scenes, we manually select a distance threshold such that all objects in the foreground are suitably visible.

**VGGT Upsampling:** Many MDE models have an internal operating resolution which is different than the input image resolution of $1280 \times 720$. With the exception of VGGT, each model specifies a procedure to resize its predicted depth map. For VGGT, which outputs the depth map at a lower resolution, we follow the upsampling procedure used by DepthAnything. Specifically, we apply bilinear upsampling with corner alignment.

## A.1  Computational Resources

We evaluate all models using jobs with 20GB of memory and either one NVIDIA RTX 3090 GPU or one NVIDIA RTX 2080 GPU. Most models take under 1 second to run for each image on the 3090 GPU, with the exceptions of Marigold and GeoWizard.

# B  Full Scene Analysis

In Tables 4 and 5 we provide detailed results when evaluating models on the full scene. In this setting, MoGe performs the best, followed by UniDepthV2 and DepthPro. As when evaluating on individual objects, UniDepthV2 achieves the lowest accuracy instability. When comparing between variations, we would expect the accuracy instability to be correlated with the pixel-level change. For example, the errors on variations which exclusively affect the object of interest (e.g. object material swap,

object translation, non-rigid object deformation) should be stable when evaluated on the full scene. We observe this to be the case. Notably, the lighting variation also has a low error and instability, giving further evidence that many frontier MDE models are robust to lighting changes.

When computing self-consistency, we restrict ourselves to variations that preserve the geometry at every pixel. This excludes Camera Roll and Object Resizing. Unlike the case of a single object, UniDepthV2 is not very consistent compared to other models. Instead, ZoeDepth and DepthPro produce the most self-consistent full scene predictions.

| Perturbation Type | Metric | MiDaS | DAv1 | DAv2 | ZoeD | UniDV2 | Metric3DV2 | Marigold | DepthPro | MoGe | GeoWizard | VGGT | Avg |
|---|---|---|---|---|---|---|---|---|---|---|---|---|---|
| Cam Dolly Zoom | Avg. Err. (↓) | 7.78 | 7.12 | 6.29 | 11.84 | 7.05 | 8.23 | 11.16 | 7.75 | **5.80** | 9.85 | 8.78 | 8.33 |
| | Acc. (In)Stab. (↓) | 5.14 | 5.82 | 4.46 | 6.14 | 3.66 | 4.97 | 5.69 | 3.14 | **2.59** | 4.65 | 3.79 | 4.55 |
| Cam Roll | Avg. Err. (↓) | 16.92 | 15.17 | 13.21 | 22.42 | 10.06 | 12.38 | 17.55 | 11.89 | **9.94** | 17.85 | 12.83 | 14.56 |
| | Acc. (In)Stab. (↓) | 9.04 | 8.41 | 6.98 | 7.21 | **3.96** | 4.79 | 7.23 | 4.52 | 4.04 | 7.17 | 4.79 | 6.19 |
| Cam Pan/Tilt | Avg. Err. (↓) | 9.35 | 8.16 | 7.85 | 11.50 | **5.79** | 6.77 | 11.89 | 7.18 | 9.10 | 19.38 | 7.59 | 9.51 |
| | Acc. (In)Stab. (↓) | 4.14 | 3.73 | 3.82 | 5.45 | **2.26** | 2.56 | 6.40 | 3.40 | 12.21 | 24.87 | 3.11 | 6.54 |
| Obj Material Swap | Avg. Err. (↓) | 8.62 | 7.91 | 7.49 | 10.35 | 6.62 | 7.17 | 10.74 | 7.02 | **5.23** | 9.88 | 8.44 | 8.13 |
| | Acc. (In)Stab. (↓) | 0.44 | 0.31 | **0.23** | 0.40 | 0.27 | 0.33 | 0.68 | 0.31 | 0.25 | 0.52 | 0.41 | 0.38 |
| Scene Material Swap | Avg. Err. (↓) | 9.78 | 8.29 | 7.92 | 11.75 | 6.75 | 7.69 | 12.34 | 8.35 | **5.38** | 10.55 | 8.91 | 8.88 |
| | Acc. (In)Stab. (↓) | 1.79 | 0.98 | **0.91** | 1.91 | 1.22 | 1.78 | 3.14 | 2.89 | 1.23 | 1.77 | 1.38 | 1.73 |
| Lighting | Avg. Err. (↓) | 8.72 | 7.88 | 7.48 | 10.35 | 6.50 | 7.06 | 10.97 | 7.81 | **5.03** | 9.71 | 8.06 | 8.14 |
| | Acc. (In)Stab. (↓) | 0.81 | 0.56 | **0.49** | 0.92 | 0.68 | 0.69 | 2.52 | 3.43 | 0.88 | 1.06 | 0.93 | 1.18 |
| Obj Rotation | Avg. Err. (↓) | 8.57 | 7.97 | **7.50** | 13.36 | 9.76 | 10.20 | 13.82 | 10.22 | 8.37 | 13.21 | 11.36 | 10.39 |
| | Acc. (In)Stab. (↓) | 0.38 | 0.25 | **0.23** | 1.93 | 1.77 | 1.75 | 2.20 | 1.98 | 1.81 | 2.38 | 1.83 | 1.50 |
| Obj Resizing | Avg. Err. (↓) | 9.55 | 7.84 | 7.18 | 11.83 | 6.99 | 7.34 | 11.01 | 7.07 | **5.99** | 10.00 | 8.56 | 8.49 |
| | Acc. (In)Stab. (↓) | 6.04 | 3.80 | 3.28 | 3.95 | 2.63 | 1.26 | 2.65 | **1.16** | 2.37 | 3.79 | 3.09 | 3.09 |
| Obj Translation | Avg. Err. (↓) | 8.56 | 7.94 | 7.50 | 10.34 | 6.69 | 7.22 | 10.62 | 6.77 | **5.40** | 9.76 | 8.55 | 8.12 |
| | Acc. (In)Stab. (↓) | 0.61 | 0.42 | **0.33** | 0.57 | 0.35 | 0.42 | 0.74 | 0.47 | 0.45 | 0.80 | 0.53 | 0.52 |
| Obj Occlusion | Avg. Err. (↓) | 9.38 | 8.50 | 8.06 | 11.49 | 7.37 | 7.46 | 11.75 | 7.35 | **5.58** | 10.60 | 9.15 | 8.79 |
| | Acc. (In)Stab. (↓) | 0.63 | 0.41 | **0.34** | 0.66 | 0.43 | 0.53 | 0.97 | 0.47 | 0.44 | 0.76 | 0.55 | 0.56 |
| Non-Rigid Obj Deformation | Avg. Err. (↓) | 8.68 | 7.94 | 7.45 | 10.29 | 6.70 | 7.20 | 10.70 | 7.01 | **5.27** | 9.79 | 8.26 | 8.12 |
| | Acc. (In)Stab. (↓) | 0.36 | 0.25 | **0.21** | 0.40 | 0.25 | 0.30 | 0.63 | 0.37 | 0.25 | 0.56 | 0.32 | 0.35 |
| Average | Avg. Err. (↓) | 9.63 | 8.61 | 7.99 | 12.32 | 7.30 | 8.06 | 12.05 | 8.04 | **6.46** | 11.87 | 9.13 | 9.22 |
| | Acc. (In)Stab. (↓) | 2.67 | 2.27 | 1.93 | 2.69 | **1.59** | 1.76 | 2.99 | 2.01 | 2.41 | 4.39 | 1.89 | 2.42 |

Table 4: Performance of depth estimation models across different perturbations, measured in the error and stability of the AbsRel metric. Full scene evaluation. Best results are highlighted in bold.

| Perturbation Type | Metric | ZoeD | UniDV2 | Metric3DV2 | DepthPro | MoGe | VGGT | Avg |
|---|---|---|---|---|---|---|---|---|
| Obj Material Swap | Self Con. (↓) | 1.58 | 2.73 | 3.84 | 1.72 | **1.16** | 2.35 | 2.23 |
| Scene Material Swap | Self Con. (↓) | 6.24 | 10.86 | 14.64 | **5.58** | 6.11 | 6.70 | 8.35 |
| Lighting | Self Con. (↓) | **4.06** | 12.33 | 10.72 | 4.82 | 5.92 | 5.61 | 7.24 |
| Average | Self Con. (↓) | **3.96** | 8.64 | 9.73 | 4.04 | 4.40 | 4.88 | 5.94 |

Table 5: Self-consistency of the AbsRel metric across depth estimation models and perturbations. Full scene evaluation. Best results are highlighted in bold.

## C Real World Validation Experiment

As discussed previously, synthetic data uniquely allows us to precisely evaluate robustness across many scenes and perturbation types. Since humans can accurately judge the depth of synthetic data, it functions well as a diagonstic probe – we can identify the manners in which existing models lack robustness. However, we hope that good performance on our dataset correlates with robustness in real-world scenarios.

To better understand this, we collect a small dataset using a mug as the object to which we apply the perturbations of Camera Pan/Tilt, Camera Roll, Lighting, Occlusion, and Object Rotation. We collect at least 15 images for each perturbation. To ensure we obtain the highly accurate ground truth depth needed for object evaluation, we calculate the depth by scanning the 3D shape of the object using an Einscan SP structured light 3D scanner. For each image, we manually annotate the 3D pose of the mug and the object mask so that the depth can be calculated using the 3D model and a calibrated camera. Figure 9 shows example images and depth maps.

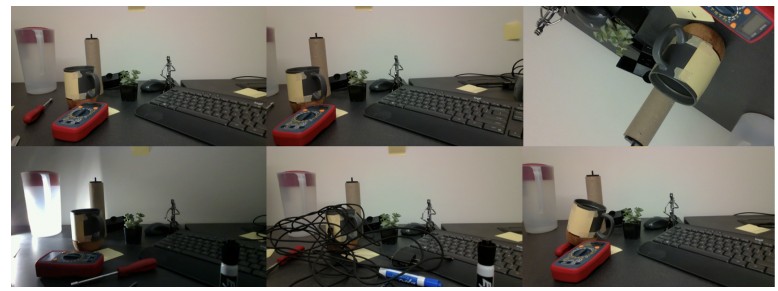 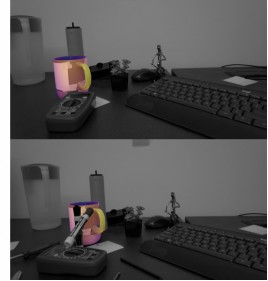

(a) Example images. Top: base, Camera Pan/Tilt, Camera Roll. Bottom: Lighting, Occlusion, Object Rotation

(b) Depth maps overlaid on images

Figure 9: Real World Experiment

We display the results in Table 6. We find similar though not identical trends in terms of models' average performance, with UniDepthV2, DepthAnythingV2, and DepthPro remaining as 3 of the top 4 models by average error. We find that occlusion remains a difficult task and many models remain fairly robust to changes in lighting. The largest difference compared to our main experiment is that the camera roll variation does not pose as significant a challenge. This may be explained by the fact that limited setting of a small desktop scene is significantly different from a larger indoor or outdoor scene.

| Perturbation Type | Metric | MiDaS | DAv1 | DAv2 | ZoeD | UniDV2 | Metric3DV2 | Marigold | DepthPro | MoGe | GeoWizard | VGGT | Avg |
|---|---|---|---|---|---|---|---|---|---|---|---|---|---|
| Cam Roll | Avg. Err. (↓) | 1.96 | 1.92 | 1.23 | 2.05 | 1.16 | 1.16 | 2.11 | 1.43 | 1.80 | 1.70 | **1.03** | 1.60 |
| | Acc. (In)Stab. (↓) | 0.38 | 0.40 | 0.34 | 0.54 | 0.31 | 0.27 | 0.42 | 0.39 | 0.42 | 0.34 | **0.16** | 0.36 |
| Cam Pan/Tilt | Avg. Err. (↓) | 2.30 | 2.02 | 1.55 | 2.33 | **1.26** | 1.66 | 2.26 | 1.58 | 1.72 | 2.18 | 1.79 | 1.88 |
| | Acc. (In)Stab. (↓) | 0.18 | 0.20 | 0.31 | 0.21 | 0.27 | 0.28 | **0.15** | 0.36 | 0.33 | 0.21 | 0.30 | 0.26 |
| Lighting | Avg. Err. (↓) | 2.63 | 2.33 | 1.62 | 2.60 | **1.22** | 1.62 | 2.45 | 1.66 | 1.81 | 2.15 | 1.70 | 1.98 |
| | Acc. (In)Stab. (↓) | **0.20** | 0.25 | 0.26 | 0.29 | 0.45 | 0.34 | 0.30 | 0.25 | 0.37 | 0.34 | 0.31 | 0.31 |
| Obj Rotation | Avg. Err. (↓) | 2.38 | 2.15 | 1.73 | 2.36 | **1.65** | 1.82 | 2.49 | 1.76 | 2.00 | 2.36 | 2.00 | 2.06 |
| | Acc. (In)Stab. (↓) | 0.60 | 0.52 | 0.41 | 0.54 | 0.37 | 0.38 | 0.55 | **0.36** | 0.46 | 0.49 | 0.42 | 0.46 |
| Obj Occlusion | Avg. Err. (↓) | 2.44 | 2.28 | 1.68 | 2.58 | **1.64** | 1.86 | 2.54 | 1.70 | 1.99 | 2.48 | 2.13 | 2.12 |
| | Acc. (In)Stab. (↓) | 0.46 | 0.35 | 0.41 | 0.42 | 0.44 | 0.39 | 0.32 | **0.26** | 0.34 | 0.36 | 0.49 | 0.38 |
| Average | Avg. Err. (↓) | 2.34 | 2.14 | 1.56 | 2.38 | **1.38** | 1.63 | 2.37 | 1.63 | 1.86 | 2.17 | 1.73 | 1.93 |
| | Acc. (In)Stab. (↓) | 0.36 | 0.34 | 0.35 | 0.40 | 0.37 | 0.33 | 0.35 | **0.32** | 0.39 | 0.35 | 0.34 | 0.35 |

Table 6: Performance of depth estimation models across different perturbations, measured in the error and stability of the AbsRel metric. Best results are highlighted in bold.

# D   Additional Analyses

## D.1   Out of Distribution Scenes

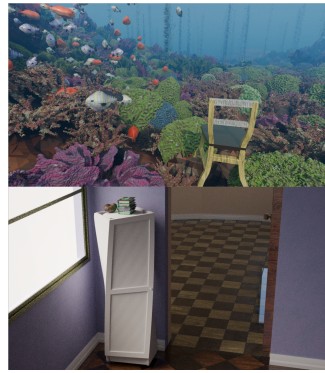 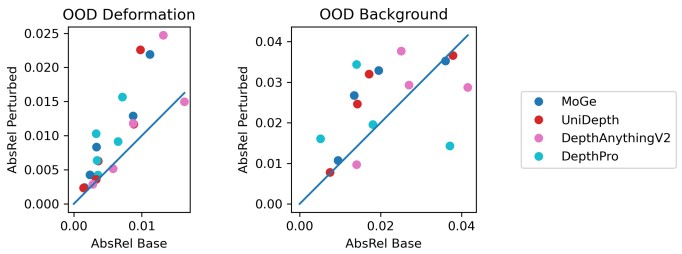

(a) Example OOD Background Swap (top) and OOD Deformation (bottom)

(b) Out of Distribution Performance of Top Performing Models. OOD Deformation is shown for desks and cabinets; OOD Background Swap is shown for chairs. Points above the blue line have higher error on the perturbed scene than the base scene.

| Metric | ZoeDepth | MiDaS | Marigold | DAv2 | DAv1 | MoGe |
|---|---|---|---|---|---|---|
| Internal Resolution | $512 \times 384$ | $512 \times 512$ | $768 \times 432$ | $920 \times 518$ | $920 \times 518$ | $530 \times 942$ |
| % Original Pixels | 21% | 28% | 36% | 52% | 52% | 54% |

| Metric | UniDV2 | Metric3DV2 | DepthPro | GeoWizard | VGGT |
|---|---|---|---|---|---|
| Internal Resolution | $588 \times 1036$ | $1064 \times 616$ | $1536 \times 1536$ | $768 \times 432$ | $518 \times 294$ |
| % Original Pixels | 66% | 71% | 256% | 36% | 17% |

Table 7: Internal resolution used by each depth estimation model and the percentage of original pixels retained (original size: $1280 \times 720$).

We manually create out of distribution scenes of two kinds: deforming the object's geometry in an unexpected manner ("OOD Deformation") and placing the object in an unrelated location ("OOD Background Swap"). In figure 10b we select top performing models DepthPro, UniDepthV2, and MoGe to compare their performance on the base scene and the OOD scene. We observe that geometry deformations have a proportionally larger impact on the error than background swaps. However, in the examples we tested the models remain decently robust to even these extreme perturbations.

## D.2 The Importance of Edge Predictions

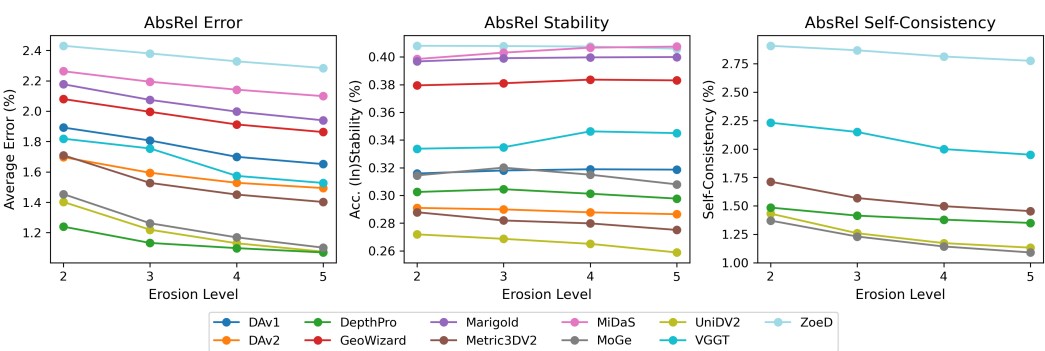

Figure 11: Impact of object mask erosion on model performance.

One factor with a significant impact on model performance is the precision of the model's edge predictions. Many models exhibit "bleeding edges" behavior where the depth prediction interpolates between the foreground and background object. This creates 3D point predictions at locations which are implausible. For objects such as certain chairs or a desk with thin legs, these edge prediction errors can account for a substantial portion of the total error. Moreover, this can lead to poor alignment when computing the scale and shift with least squares minimization. In figure 11, we perform the same evaluation while ignoring pixels close to the boundary of the object mask. The "erosion level" refers to the size of the radius (in pixels) we use to shrink the object mask. Notably, we observe that as we increase the erosion level, MoGe and UniDepthV2 approach the error score of DepthPro. While self-consistency improves as the erosion level increases, the trend is weaker for error stability. This suggests that improving edge prediction accuracy is important for average error, but may not impact the robustness of a model.

## D.3 Object Size Analysis

Closely related to predicting sharp edges is the internal resolution used by the model. Models which operate at a lower resolution may have difficulties perceiving the detail necessary to predict sharp edges and small objects. Here, we analyze whether the error on small objects explains the average error gap between DepthPro and other methods. We classify object size using the pixel thresholds from the COCO [22] small, medium, and large categories appropriately scaled to our image resolution. Our dataset consists of 32 large objects, 8 medium objects, and 0 small objects. In table 8 we compare the models' scores on the full dataset and the subset of the dataset with "large" objects. We display the internal operating resolution of each model in table 7. Notably, many of the models which operate at fairly low resolutions (e.g. VGGT, ZoeDepth, GeoWizard) display a performance degradation

when removing the small objects. That is, these models performed better on smaller objects than larger objects. Thus, the gap in performance cannot be explained by small objects lacking detail when the image is downsampled.

| Metric | MiDaS | DAv1 | DAv2 | ZoeDepth | UniDV2 | Metric3DV2 | Marigold | DepthPro | MoGe | GeoWizard | VGGT |
|---|---|---|---|---|---|---|---|---|---|---|---|
| Original Avg. Error | 2.26 | 1.89 | 1.70 | 2.43 | 1.40 | 1.71 | 2.18 | 1.24 | 1.45 | 2.08 | 1.82 |
| Large Objects Avg. Error | 2.44 | 1.99 | 1.78 | 2.61 | 1.39 | 1.76 | 2.28 | 1.27 | 1.44 | 2.15 | 1.88 |
| % Degradation | 7.70% | 5.22% | 5.13% | 7.31% | -0.76% | 3.10% | 4.90% | 2.37% | -0.76% | 3.22% | 3.20% |
| Original Acc. (In)Stability | 0.40 | 0.32 | 0.29 | 0.41 | 0.27 | 0.29 | 0.40 | 0.30 | 0.31 | 0.38 | 0.33 |
| Large Objects Acc. (In)Stability | 0.43 | 0.34 | 0.31 | 0.44 | 0.28 | 0.29 | 0.43 | 0.32 | 0.31 | 0.40 | 0.35 |
| % Degradation | 7.27% | 7.13% | 6.56% | 6.84% | 3.62% | -0.71% | 7.32% | 4.79% | -0.02% | 4.17% | 5.14% |

Table 8: Comparison of model performance on original data vs. large objects only (excluding small objects). Degradation shows the percentage increase in error/instability when excluding small objects.

### D.4 Combined Variations

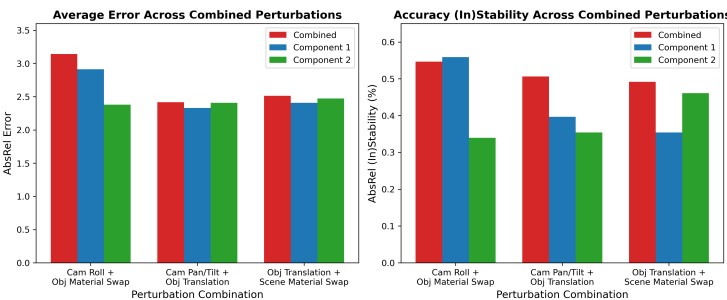

Figure 12: Combined variations: each group of bars displays statistics for the combined perturbation and its two subcomponents. For example, the leftmost red bar displays the error for the combined Camera Roll and Object Material Swap perturbation, the blue bar shows the Camera Roll perturbation, and the green bar shows the object material swap perturbation

The majority of our variations alter the scene such that the perturbed version is also a physically plausible scene. Moreover, estimating the depth of the perturbed scene should be equally easy as estimating the depth of the base scene. Therefore, we expect that combining in-distribution perturbations should not have a large impact on the error. To test this, we generate combined perturbations for the chairs and desks objects consisting of (1) Camera Roll and Object Material Swap (2) Camera Pan/Tilt and Object Translation (3) Object Translation and Scene Material Swap. We display the results in figure 12. For these latter two perturbations, the average error of the combined perturbation was similar to each of the individual perturbations as expected. For the Camera Roll and Object Material Swap combination, the average error did increase. This may be due to the fact that models are not robust to the Camera Roll perturbation, and combining this with the material swap perturbation produces images which are further from the models' training distribution.

## E  Additional Results

We provide additional results for the $\delta_{0.125}$, RMSE, and relative depth metrics. These metrics follow standard definitions, except $\delta_{0.125}$ is stricter than the $\delta_1$ metric. This is appropriate since we are aligning the predictions to a single object.

$$\delta_{0.125} = \% \text{ of } \hat{y}_i \text{ s.t. } \max\left(\frac{\hat{y}_i}{y_i}, \frac{y_i}{\hat{y}_i}\right) < 1.25^{0.125} \qquad RMSE = \sqrt{\frac{1}{n}\sum \|y_i - \hat{y}_i\|^2}.$$

The relative depth metric describes the agreement in the depth ordering between the ground truth and predicted depth. For 10 million randomly selected pairs of pixels $i$ and $j$, we measure the percent such that $y_i < y_j$ agrees with $\hat{y}_i < \hat{y}_j$. Notably, this does not require performing alignment and thus allows us to compute the self-consistency for all models including the affine-invariant DepthAnything and MiDaS.

## E.1 Results for the Relative Depth Metric

| Perturbation Type | Metric | MiDaS | DAv1 | DAv2 | ZoeD | UniDV2 | Metric3DV2 | Marigold | DepthPro | MoGe | GeoWizard | VGGT | Avg |
|---|---|---|---|---|---|---|---|---|---|---|---|---|---|
| Cam Dolly Zoom | Avg. Err. (↑) | 78.29 | 84.09 | 84.13 | 74.57 | 87.04 | 83.27 | 77.53 | **87.54** | 87.50 | 77.28 | 83.06 | 82.21 |
| | Acc. (In)Stab. (↓) | 3.72 | 2.47 | **1.77** | 4.06 | 1.80 | 2.58 | 3.88 | 2.66 | 2.22 | 3.96 | 2.28 | 2.86 |
| Cam Roll | Avg. Err. (↑) | 70.25 | 76.38 | 78.31 | 65.95 | **80.08** | 78.63 | 68.70 | 78.59 | 79.53 | 70.00 | 76.16 | 74.78 |
| | Acc. (In)Stab. (↓) | 7.93 | 6.48 | 6.33 | 8.72 | 5.56 | 5.87 | 7.95 | 8.36 | 5.47 | 7.69 | **5.38** | 6.88 |
| Cam Pan/Tilt | Avg. Err. (↑) | 77.19 | 83.12 | 83.35 | 75.01 | 87.07 | 82.19 | 76.64 | **87.29** | 86.36 | 77.26 | 83.37 | 81.71 |
| | Acc. (In)Stab. (↓) | 5.46 | 4.49 | 4.83 | 6.89 | **3.50** | 4.36 | 5.62 | 4.85 | 4.49 | 5.42 | 4.11 | 4.91 |
| Obj Material Swap | Avg. Err. (↑) | 75.24 | 83.27 | 82.79 | 72.62 | 85.51 | 82.36 | 75.78 | 85.66 | **86.33** | 76.40 | 82.74 | 80.79 |
| | Acc. (In)Stab. (↓) | 5.42 | 3.84 | **3.06** | 5.87 | 3.22 | 3.70 | 5.29 | 3.45 | 3.80 | 5.37 | 4.83 | 4.35 |
| Scene Material Swap | Avg. Err. (↑) | 75.76 | 82.77 | 82.70 | 74.70 | **85.44** | 82.04 | 75.91 | 84.05 | 85.21 | 76.51 | 82.35 | 80.68 |
| | Acc. (In)Stab. (↓) | 5.63 | 4.78 | **4.03** | 6.62 | 4.21 | 4.46 | 5.64 | 6.12 | 5.04 | 5.46 | 5.49 | 5.23 |
| Lighting | Avg. Err. (↑) | 75.94 | 82.31 | 82.28 | 73.24 | 85.50 | 82.17 | 75.85 | **86.16** | 85.25 | 75.96 | 82.57 | 80.66 |
| | Acc. (In)Stab. (↓) | 3.60 | 2.72 | **2.05** | 4.65 | 2.42 | 2.75 | 4.48 | 3.13 | 2.98 | 3.94 | 3.34 | 3.28 |
| Obj Rotation | Avg. Err. (↑) | 77.10 | 83.23 | 83.12 | 75.54 | 85.88 | 82.24 | 76.02 | **86.49** | 85.95 | 76.04 | 82.11 | 81.25 |
| | Acc. (In)Stab. (↓) | 4.40 | 3.60 | 3.72 | 5.58 | **3.19** | 3.86 | 4.95 | 3.73 | 4.23 | 5.13 | 4.22 | 4.24 |
| Obj Resizing | Avg. Err. (↑) | 77.87 | 83.14 | 83.12 | 75.14 | 86.58 | 81.91 | 77.37 | **87.21** | 86.63 | 76.89 | 83.10 | 81.72 |
| | Acc. (In)Stab. (↓) | 2.74 | 2.21 | **1.60** | 3.58 | 1.74 | 1.98 | 3.80 | 2.00 | 2.04 | 3.57 | 2.55 | 2.53 |
| Obj Translation | Avg. Err. (↑) | 77.01 | 83.65 | 83.36 | 74.14 | 85.87 | 81.97 | 76.31 | **86.85** | 86.16 | 76.17 | 82.52 | 81.27 |
| | Acc. (In)Stab. (↓) | 4.28 | 3.16 | 3.18 | 5.76 | **2.79** | 3.22 | 4.67 | 3.26 | 3.45 | 4.89 | 3.00 | 3.79 |
| Obj Occlusion | Avg. Err. (↑) | 73.67 | 79.91 | 80.09 | 72.50 | 82.31 | 79.52 | 71.80 | 82.31 | **83.09** | 74.29 | 79.49 | 78.09 |
| | Acc. (In)Stab. (↓) | 5.83 | 4.52 | **3.72** | 6.50 | 4.56 | 4.37 | 6.14 | 4.55 | 4.95 | 5.21 | 5.47 | 5.08 |
| Non-Rigid Obj Deformation | Avg. Err. (↑) | 78.26 | 84.12 | 83.84 | 75.30 | 86.59 | 82.94 | 77.38 | 86.82 | **87.23** | 77.35 | 83.71 | 82.14 |
| | Acc. (In)Stab. (↓) | 5.03 | 3.74 | 3.63 | 6.62 | **3.21** | 3.88 | 5.01 | 3.73 | 3.48 | 5.07 | 4.19 | 4.33 |
| Average | Avg. Err. (↑) | 76.05 | 82.36 | 82.46 | 73.52 | 85.26 | 81.75 | 75.39 | 85.36 | **85.39** | 75.83 | 81.93 | 80.48 |
| | Acc. (In)Stab. (↓) | 4.91 | 3.82 | 3.45 | 5.90 | **3.29** | 3.73 | 5.22 | 4.17 | 3.83 | 5.06 | 4.08 | 4.31 |

Table 9: Performance of depth estimation models across different perturbations, measured in the error and stability of the Rel Depth metric. Best results are highlighted in bold.

| Perturbation Type | Metric | MiDaS | DAv1 | DAv2 | ZoeD | UniDV2 | Metric3DV2 | Marigold | DepthPro | MoGe | VGGT | Avg |
|---|---|---|---|---|---|---|---|---|---|---|---|---|
| Cam Roll | Self Con. (↓) | 23.34 | 19.54 | 17.97 | 25.86 | 17.13 | 21.72 | 26.79 | 21.57 | **16.58** | 17.65 | 20.82 |
| Obj Material Swap | Self Con. (↓) | 15.63 | 10.79 | **7.56** | 18.65 | 9.30 | 13.24 | 17.88 | 10.38 | 9.32 | 13.42 | 12.62 |
| Scene Material Swap | Self Con. (↓) | 16.31 | 12.62 | **9.26** | 19.20 | 11.19 | 15.30 | 18.53 | 14.83 | 11.30 | 14.05 | 14.26 |
| Lighting | Self Con. (↓) | 10.76 | 7.86 | **5.87** | 13.56 | 7.66 | 10.45 | 14.81 | 9.20 | 8.69 | 9.88 | 9.87 |
| Obj Resizing | Self Con. (↓) | 7.98 | 6.09 | **4.69** | 10.32 | 5.33 | 7.85 | 14.28 | 6.77 | 5.48 | 7.39 | 7.62 |
| Average | Self Con. (↓) | 14.81 | 11.38 | **9.07** | 17.52 | 10.12 | 13.71 | 18.46 | 12.55 | 10.27 | 12.48 | 13.04 |

Table 10: Self-consistency of the Rel Depth metric across depth estimation models and perturbations. Best results are highlighted in bold.

## E.2 Results for metrics $\delta_{0.125}$ and RMSE for Scale and Shift Alignment

| Perturbation Type | Metric | MiDaS | DAv1 | DAv2 | ZoeD | UniDV2 | Metric3DV2 | Marigold | DepthPro | MoGe | GeoWizard | VGGT | Avg |
|---|---|---|---|---|---|---|---|---|---|---|---|---|---|
| Cam Dolly Zoom | Avg. Err. (↑) | 72.39 | 77.38 | 80.15 | 68.67 | 85.40 | 79.84 | 71.52 | **89.93** | 85.52 | 73.90 | 78.87 | 78.51 |
| | Acc. (In)Stab. (↓) | 8.43 | 7.78 | 7.79 | 8.98 | 5.66 | 6.94 | 9.98 | **4.96** | 6.09 | 10.09 | 5.68 | 7.49 |
| Cam Roll | Avg. Err. (↑) | 69.20 | 74.38 | 78.35 | 65.93 | 81.01 | 79.41 | 69.42 | **81.14** | 79.08 | 71.79 | 76.68 | 75.12 |
| | Acc. (In)Stab. (↓) | 7.78 | 7.73 | 8.18 | 6.95 | **5.49** | 6.36 | 7.69 | 8.94 | 5.69 | 7.59 | 5.84 | 7.11 |
| Cam Pan/Tilt | Avg. Err. (↑) | 74.66 | 80.08 | 82.87 | 71.20 | 87.59 | 82.88 | 75.16 | **90.81** | 87.10 | 77.46 | 80.80 | 80.96 |
| | Acc. (In)Stab. (↓) | 7.81 | 5.61 | 5.94 | 8.09 | **4.05** | 5.44 | 7.31 | 5.06 | 4.08 | 6.28 | 4.41 | 5.82 |
| Obj Material Swap | Avg. Err. (↑) | 72.91 | 80.02 | 82.74 | 71.14 | 87.16 | 81.24 | 75.55 | **90.05** | 85.38 | 77.04 | 80.61 | 80.35 |
| | Acc. (In)Stab. (↓) | 6.53 | 5.73 | 4.53 | 6.49 | 4.03 | 4.00 | 5.25 | **3.97** | 5.53 | 5.34 | 5.17 | 5.14 |
| Scene Material Swap | Avg. Err. (↑) | 73.37 | 80.09 | 83.58 | 71.63 | 87.74 | 82.52 | 75.92 | **88.33** | 85.50 | 76.73 | 80.54 | 80.54 |
| | Acc. (In)Stab. (↓) | 7.14 | 6.70 | 5.10 | 7.09 | 5.05 | **4.98** | 5.92 | 6.75 | 6.25 | 5.80 | 6.01 | 6.07 |
| Lighting | Avg. Err. (↑) | 74.19 | 79.78 | 83.50 | 72.46 | 86.47 | 81.60 | 76.57 | **91.11** | 86.35 | 77.61 | 81.81 | 81.04 |
| | Acc. (In)Stab. (↓) | 3.85 | 2.79 | 2.79 | 4.16 | **1.93** | 2.78 | 3.49 | 2.81 | 2.61 | 3.92 | 2.92 | 3.09 |
| Obj Rotation | Avg. Err. (↑) | 74.72 | 79.95 | 82.40 | 71.68 | 86.36 | 81.80 | 75.23 | **89.84** | 85.91 | 76.84 | 80.39 | 80.47 |
| | Acc. (In)Stab. (↓) | 5.55 | 5.12 | 5.52 | 5.61 | 4.34 | 4.42 | 5.70 | **4.08** | 5.32 | 5.70 | 4.43 | 5.07 |
| Obj Resizing | Avg. Err. (↑) | 74.85 | 79.83 | 82.73 | 72.55 | 86.85 | 81.24 | 76.72 | **91.20** | 86.08 | 77.05 | 80.19 | 80.84 |
| | Acc. (In)Stab. (↓) | 2.91 | 3.02 | 2.60 | 4.18 | 2.40 | 2.87 | 3.78 | **1.93** | 2.93 | 4.42 | 2.79 | 3.08 |
| Obj Translation | Avg. Err. (↑) | 74.44 | 79.46 | 82.69 | 72.11 | 86.26 | 81.46 | 75.71 | **89.71** | 86.23 | 77.44 | 80.31 | 80.53 |
| | Acc. (In)Stab. (↓) | 4.96 | 4.22 | 4.20 | 5.55 | **3.18** | 3.67 | 5.18 | 3.69 | 4.08 | 5.10 | 3.78 | 4.33 |
| Obj Occlusion | Avg. Err. (↑) | 72.38 | 76.04 | 77.79 | 71.04 | 81.98 | 78.95 | 71.67 | **83.97** | 83.08 | 74.30 | 78.39 | 77.24 |
| | Acc. (In)Stab. (↓) | 6.55 | 5.13 | **4.86** | 6.39 | 5.01 | 5.11 | 7.39 | 5.95 | 5.54 | 6.28 | 5.14 | 5.76 |
| Non-Rigid Obj Deformation | Avg. Err. (↑) | 75.70 | 80.95 | 83.42 | 72.30 | 87.30 | 82.65 | 76.50 | **89.75** | 86.44 | 78.32 | 81.26 | 81.33 |
| | Acc. (In)Stab. (↓) | 7.14 | 6.16 | 5.48 | 8.12 | **4.58** | 5.32 | 6.81 | 4.84 | 4.89 | 6.47 | 4.86 | 5.88 |
| Average | Avg. Err. (↑) | 73.53 | 78.91 | 81.84 | 70.97 | 85.83 | 81.24 | 74.54 | **88.71** | 85.15 | 76.23 | 79.99 | 79.72 |
| | Acc. (In)Stab. (↓) | 6.24 | 5.45 | 5.18 | 6.51 | **4.16** | 4.72 | 6.23 | 4.82 | 4.82 | 6.09 | 4.64 | 5.35 |

Table 11: Performance of depth estimation models across different perturbations, measured in the error and stability of the $\delta_{0.125}$ metric. Best results are highlighted in bold.

| Perturbation Type | Metric | ZoeD | UniDV2 | Metric3DV2 | DepthPro | MoGe | VGGT | Avg |
|---|---|---|---|---|---|---|---|---|
| Cam Roll | Self Con. (↓) | 41.35 | **19.79** | 27.31 | 24.83 | 21.15 | 29.36 | 27.30 |
| Obj Material Swap | Self Con. (↓) | 32.99 | 14.43 | 19.17 | **12.97** | 13.23 | 23.19 | 19.33 |
| Scene Material Swap | Self Con. (↓) | 34.94 | 16.79 | 20.81 | 18.19 | **15.91** | 24.42 | 21.84 |
| Lighting | Self Con. (↓) | 23.10 | 9.86 | 13.41 | 9.97 | **8.16** | 17.07 | 13.59 |
| Obj Resizing | Self Con. (↓) | 16.52 | 6.55 | 9.55 | **6.40** | 6.61 | 14.86 | 10.08 |
| Average | Self Con. (↓) | 29.78 | 13.48 | 18.05 | 14.47 | **13.01** | 21.78 | 18.43 |

Table 12: Self-consistency of the $\delta_{0.125}$ metric across depth estimation models and perturbations. Best results are highlighted in bold.

| Perturbation Type | Metric | MiDaS | DAv1 | DAv2 | ZoeD | UniDV2 | Metric3DV2 | Marigold | DepthPro | MoGe | GeoWizard | VGGT | Avg |
|---|---|---|---|---|---|---|---|---|---|---|---|---|---|
| Cam Dolly Zoom | Avg. Err. (↓) | 13.10 | 12.62 | 11.47 | 13.81 | 11.00 | 11.64 | 14.37 | **8.69** | 11.10 | 13.51 | 12.92 | 12.20 |
| | Acc. (In)Stab. (↓) | 1.64 | 1.20 | **1.08** | 1.79 | 1.12 | 1.57 | 1.51 | 1.57 | 1.26 | 1.70 | 1.25 | 1.43 |
| Cam Roll | Avg. Err. (↓) | 17.29 | 16.19 | 14.03 | 18.49 | 13.96 | 14.36 | 18.21 | **13.34** | 14.54 | 17.43 | 15.56 | 15.76 |
| | Acc. (In)Stab. (↓) | 2.88 | 2.43 | 2.80 | 2.68 | **1.87** | 2.59 | 2.65 | 3.03 | 2.09 | 2.47 | 2.11 | 2.51 |
| Cam Pan/Tilt | Avg. Err. (↓) | 13.57 | 12.81 | 11.54 | 14.54 | 10.86 | 11.65 | 14.74 | **9.03** | 11.12 | 13.94 | 12.60 | 12.40 |
| | Acc. (In)Stab. (↓) | 2.65 | **1.89** | 1.98 | 2.88 | 1.93 | 2.34 | 2.57 | 2.29 | 2.03 | 2.54 | 2.27 | 2.31 |
| Obj Material Swap | Avg. Err. (↓) | 14.50 | 13.07 | 12.15 | 14.65 | 11.48 | 12.78 | 14.97 | **9.32** | 12.10 | 14.03 | 13.53 | 12.96 |
| | Acc. (In)Stab. (↓) | 2.19 | 1.76 | 1.43 | 2.21 | **1.38** | 1.54 | 1.62 | 1.56 | 1.84 | 1.68 | 1.69 | 1.72 |
| Scene Material Swap | Avg. Err. (↓) | 13.57 | 12.64 | 11.35 | 13.97 | 10.89 | 11.69 | 14.03 | **9.56** | 11.59 | 13.65 | 12.88 | 12.35 |
| | Acc. (In)Stab. (↓) | 2.25 | 2.08 | 1.81 | 2.41 | **1.69** | 1.94 | 1.94 | 2.36 | 2.23 | 2.06 | 2.16 | 2.09 |
| Lighting | Avg. Err. (↓) | 13.72 | 13.23 | 11.74 | 14.31 | 11.55 | 12.06 | 14.51 | **9.40** | 11.49 | 13.99 | 13.01 | 12.64 |
| | Acc. (In)Stab. (↓) | 1.43 | 0.88 | 0.98 | 1.56 | **0.81** | 1.13 | 1.20 | 1.44 | 1.14 | 1.40 | 1.23 | 1.20 |
| Obj Rotation | Avg. Err. (↓) | 13.56 | 12.90 | 11.94 | 14.51 | 11.59 | 12.21 | 14.68 | **9.26** | 11.69 | 13.95 | 13.43 | 12.70 |
| | Acc. (In)Stab. (↓) | 2.25 | 1.97 | 1.92 | 2.28 | **1.82** | 1.94 | 2.21 | 1.83 | 2.27 | 2.37 | 2.07 | 2.09 |
| Obj Resizing | Avg. Err. (↓) | 13.29 | 12.67 | 11.75 | 13.67 | 11.21 | 11.91 | 14.20 | **8.90** | 11.59 | 13.51 | 13.01 | 12.34 |
| | Acc. (In)Stab. (↓) | 2.11 | 1.94 | 1.80 | 2.35 | 1.97 | 1.95 | 2.40 | **1.63** | 1.99 | 2.46 | 2.07 | 2.06 |
| Obj Translation | Avg. Err. (↓) | 13.91 | 13.24 | 12.09 | 14.63 | 11.85 | 12.67 | 14.84 | **9.41** | 12.11 | 14.10 | 13.60 | 12.95 |
| | Acc. (In)Stab. (↓) | 1.81 | 1.39 | 1.32 | 2.02 | **1.21** | 1.44 | 1.63 | 1.53 | 1.69 | 1.83 | 1.47 | 1.58 |
| Obj Occlusion | Avg. Err. (↓) | 14.04 | 13.55 | 12.87 | 14.67 | 12.84 | 13.32 | 15.83 | **11.40** | 11.72 | 15.02 | 13.52 | 13.53 |
| | Acc. (In)Stab. (↓) | 2.55 | 2.04 | **1.85** | 2.66 | 1.94 | 1.96 | 2.54 | 2.05 | 2.35 | 2.26 | 2.52 | 2.25 |
| Non-Rigid Obj Deformation | Avg. Err. (↓) | 13.68 | 13.02 | 11.97 | 14.40 | 11.55 | 12.21 | 14.60 | **9.20** | 11.60 | 13.82 | 13.32 | 12.67 |
| | Acc. (In)Stab. (↓) | 3.54 | 2.82 | 2.44 | 3.30 | 2.53 | **2.40** | 3.07 | 2.55 | 2.68 | 3.14 | 2.94 | 2.86 |
| Average | Avg. Err. (↓) | 14.02 | 13.27 | 12.08 | 14.69 | 11.71 | 12.41 | 15.00 | **9.77** | 11.88 | 14.27 | 13.40 | 12.95 |
| | Acc. (In)Stab. (↓) | 2.30 | 1.85 | 1.77 | 2.38 | **1.66** | 1.89 | 2.12 | 1.99 | 1.96 | 2.17 | 1.98 | 2.01 |

Table 13: Performance of depth estimation models across different perturbations, measured in the error and stability of the RMSE metric (in cm). Best results are highlighted in bold.

| Perturbation Type | Metric | ZoeD | UniDV2 | Metric3DV2 | DepthPro | MoGe | VGGT | Avg |
|---|---|---|---|---|---|---|---|---|
| Cam Roll | Self Con. (↓) | 5.68 | 4.11 | 4.40 | **3.06** | 3.71 | 5.93 | 4.48 |
| Obj Material Swap | Self Con. (↓) | 4.23 | 4.30 | 4.02 | **1.88** | 3.10 | 3.95 | 3.58 |
| Scene Material Swap | Self Con. (↓) | 4.04 | 3.78 | 3.44 | **2.29** | 3.38 | 4.43 | 3.56 |
| Lighting | Self Con. (↓) | 2.94 | 3.93 | 3.74 | **1.70** | 2.57 | 3.51 | 3.07 |
| Obj Resizing | Self Con. (↓) | 2.52 | 3.03 | 3.15 | **1.46** | 2.19 | 3.08 | 2.57 |
| Average | Self Con. (↓) | 3.88 | 3.83 | 3.75 | **2.08** | 2.99 | 4.18 | 3.45 |

Table 14: Self-consistency of the RMSE metric (in cm) across depth estimation models and perturbations. Best results are highlighted in bold.

## E.3 Results for metrics AbsRel, $\delta_{0.125}$ and RMSE for Scale-Only Alignment

| Perturbation Type | Metric | ZoeD | UniDV2 | Metric3DV2 | DepthPro | MoGe | VGGT | Avg |
|---|---|---|---|---|---|---|---|---|
| Cam Dolly Zoom | Avg. Err. ($\downarrow$) | 5.53 | 3.46 | 3.99 | **2.21** | 2.46 | 3.64 | 3.55 |
| | Acc. (In)Stab. ($\downarrow$) | 1.70 | 1.26 | 1.33 | 0.79 | **0.75** | 1.39 | 1.20 |
| Cam Roll | Avg. Err. ($\downarrow$) | 5.34 | **3.34** | 3.95 | 3.48 | 3.53 | 5.49 | 4.19 |
| | Acc. (In)Stab. ($\downarrow$) | 2.07 | 1.54 | 1.55 | 1.56 | **1.48** | 2.27 | 1.75 |
| Cam Pan/Tilt | Avg. Err. ($\downarrow$) | 5.48 | 2.36 | 3.07 | **2.07** | 2.59 | 3.78 | 3.23 |
| | Acc. (In)Stab. ($\downarrow$) | 1.77 | 0.88 | 1.31 | **0.70** | 0.75 | 1.37 | 1.13 |
| Obj Material Swap | Avg. Err. ($\downarrow$) | 5.65 | 2.59 | 2.83 | **1.91** | 2.33 | 3.80 | 3.18 |
| | Acc. (In)Stab. ($\downarrow$) | 2.21 | 1.14 | 0.85 | **0.71** | 0.85 | 2.32 | 1.35 |
| Scene Material Swap | Avg. Err. ($\downarrow$) | 6.23 | **2.34** | 2.56 | 2.57 | 2.86 | 3.87 | 3.40 |
| | Acc. (In)Stab. ($\downarrow$) | 2.40 | 1.38 | **1.04** | 1.57 | 1.77 | 2.91 | 1.85 |
| Lighting | Avg. Err. ($\downarrow$) | 5.58 | 2.39 | 3.09 | **1.93** | 2.45 | 3.42 | 3.14 |
| | Acc. (In)Stab. ($\downarrow$) | 1.44 | 0.91 | 0.87 | 0.69 | **0.66** | 1.42 | 1.00 |
| Obj Rotation | Avg. Err. ($\downarrow$) | 5.44 | 2.99 | 3.65 | **2.21** | 2.53 | 3.48 | 3.38 |
| | Acc. (In)Stab. ($\downarrow$) | 1.42 | **0.65** | 0.90 | 0.68 | 0.73 | 1.34 | 0.95 |
| Obj Resizing | Avg. Err. ($\downarrow$) | 5.58 | 3.05 | 3.45 | **2.19** | 2.48 | 3.63 | 3.40 |
| | Acc. (In)Stab. ($\downarrow$) | 1.01 | 0.57 | 0.64 | 0.61 | **0.40** | 1.28 | 0.75 |
| Obj Translation | Avg. Err. ($\downarrow$) | 5.04 | 3.23 | 3.91 | **2.20** | 2.50 | 3.46 | 3.39 |
| | Acc. (In)Stab. ($\downarrow$) | 1.35 | 1.07 | 1.55 | 0.68 | **0.68** | 1.23 | 1.09 |
| Obj Occlusion | Avg. Err. ($\downarrow$) | 5.13 | 2.50 | 2.86 | **2.19** | 2.32 | 3.47 | 3.08 |
| | Acc. (In)Stab. ($\downarrow$) | 1.46 | 0.84 | 0.83 | **0.69** | 0.82 | 1.33 | 1.00 |
| Non-Rigid Obj Deformation | Avg. Err. ($\downarrow$) | 4.91 | 2.76 | 3.46 | **2.08** | 2.33 | 3.30 | 3.14 |
| | Acc. (In)Stab. ($\downarrow$) | 1.51 | 1.01 | 1.02 | 0.72 | **0.72** | 1.23 | 1.03 |
| Average | Avg. Err. ($\downarrow$) | 5.45 | 2.82 | 3.35 | **2.28** | 2.58 | 3.76 | 3.37 |
| | Acc. (In)Stab. ($\downarrow$) | 1.67 | 1.02 | 1.08 | **0.85** | 0.87 | 1.65 | 1.19 |

Table 15: Performance of depth estimation models across different perturbations, measured in the error and stability of the AbsRel metric. The scale of each prediction was aligned for evaluation. Best results are highlighted in bold.

| Perturbation Type | Metric | ZoeD | UniDV2 | Metric3DV2 | DepthPro | MoGe | VGGT | Avg |
|---|---|---|---|---|---|---|---|---|
| Cam Roll | Self Con. ($\downarrow$) | 4.95 | **2.69** | 3.48 | 3.35 | 3.17 | 4.91 | 3.76 |
| Obj Material Swap | Self Con. ($\downarrow$) | 4.48 | 2.09 | 2.12 | 1.81 | **1.72** | 3.71 | 2.66 |
| Scene Material Swap | Self Con. ($\downarrow$) | 4.75 | **2.51** | 2.52 | 2.66 | 2.80 | 4.45 | 3.28 |
| Lighting | Self Con. ($\downarrow$) | 2.90 | **1.48** | 1.72 | 1.54 | 1.52 | 2.53 | 1.95 |
| Obj Resizing | Self Con. ($\downarrow$) | 2.14 | 1.07 | 1.48 | 1.16 | **0.99** | 2.36 | 1.53 |
| Average | Self Con. ($\downarrow$) | 3.84 | **1.97** | 2.26 | 2.11 | 2.04 | 3.59 | 2.64 |

Table 16: Self-consistency of the AbsRel metric across depth estimation models and perturbations. The scale of each prediction was aligned for evaluation. Best results are highlighted in bold.

| Perturbation Type | Metric | ZoeD | UniDV2 | Metric3DV2 | DepthPro | MoGe | VGGT | Avg |
|---|---|---|---|---|---|---|---|---|
| Cam Dolly Zoom | Avg. Err. (↑) | 47.11 | 73.46 | 65.17 | **77.37** | 75.88 | 69.26 | 68.04 |
| | Acc. (In)Stab. (↓) | 12.40 | 8.89 | 9.99 | 9.39 | 12.17 | **8.33** | 10.20 |
| Cam Roll | Avg. Err. (↑) | 42.78 | **66.29** | 60.13 | 59.77 | 65.97 | 58.48 | 58.90 |
| | Acc. (In)Stab. (↓) | 14.79 | 14.68 | 13.58 | 18.09 | 14.50 | **12.31** | 14.66 |
| Cam Pan/Tilt | Avg. Err. (↑) | 46.80 | 76.99 | 69.41 | **79.25** | 74.46 | 70.44 | 69.56 |
| | Acc. (In)Stab. (↓) | 13.04 | **7.84** | 10.37 | 9.76 | 10.26 | 8.60 | 9.98 |
| Obj Material Swap | Avg. Err. (↑) | 43.78 | 76.95 | 71.35 | **80.72** | 76.37 | 69.52 | 69.78 |
| | Acc. (In)Stab. (↓) | 13.10 | **9.09** | 9.48 | 9.49 | 11.14 | 13.32 | 10.94 |
| Scene Material Swap | Avg. Err. (↑) | 43.25 | **81.15** | 72.68 | 77.35 | 73.30 | 69.50 | 69.54 |
| | Acc. (In)Stab. (↓) | 13.86 | 11.83 | **11.33** | 14.49 | 14.63 | 14.82 | 13.49 |
| Lighting | Avg. Err. (↑) | 46.07 | 77.08 | 68.35 | **80.33** | 75.61 | 72.01 | 69.91 |
| | Acc. (In)Stab. (↓) | 9.74 | 8.42 | **7.84** | 8.40 | 7.95 | 9.31 | 8.61 |
| Obj Rotation | Avg. Err. (↑) | 46.99 | 74.69 | 66.66 | **77.23** | 74.86 | 69.51 | 68.32 |
| | Acc. (In)Stab. (↓) | 8.75 | **6.80** | 7.67 | 8.95 | 8.71 | 8.09 | 8.16 |
| Obj Resizing | Avg. Err. (↑) | 45.95 | 76.97 | 67.77 | **79.99** | 74.97 | 70.22 | 69.31 |
| | Acc. (In)Stab. (↓) | 7.05 | **4.50** | 5.76 | 6.35 | 5.59 | 7.54 | 6.13 |
| Obj Translation | Avg. Err. (↑) | 47.55 | 73.53 | 65.95 | **77.21** | 76.01 | 69.64 | 68.32 |
| | Acc. (In)Stab. (↓) | 10.61 | **7.20** | 9.05 | 8.90 | 8.17 | 9.00 | 8.82 |
| Obj Occlusion | Avg. Err. (↑) | 48.58 | 74.52 | 68.05 | 75.94 | **75.99** | 68.67 | 68.62 |
| | Acc. (In)Stab. (↓) | 10.87 | **8.99** | 9.67 | 10.47 | 10.98 | 10.52 | 10.25 |
| Non-Rigid Obj Deformation | Avg. Err. (↑) | 47.26 | 77.02 | 70.13 | **77.69** | 77.28 | 71.72 | 70.18 |
| | Acc. (In)Stab. (↓) | 12.28 | **8.12** | 9.19 | 10.16 | 9.20 | 9.72 | 9.78 |
| Average | Avg. Err. (↑) | 46.01 | 75.33 | 67.79 | **76.62** | 74.61 | 69.00 | 68.23 |
| | Acc. (In)Stab. (↓) | 11.50 | **8.76** | 9.45 | 10.40 | 10.30 | 10.14 | 10.09 |

Table 17: Performance of depth estimation models across different perturbations, measured in the error and stability of the $\delta_{0.125}$ metric. The scale of each prediction was aligned for evaluation. Best results are highlighted in bold.

| Perturbation Type | Metric | ZoeD | UniDV2 | Metric3DV2 | DepthPro | MoGe | VGGT | Avg |
|---|---|---|---|---|---|---|---|---|
| Cam Roll | Self Con. (↓) | 53.24 | **30.78** | 40.16 | 40.54 | 32.55 | 40.02 | 39.55 |
| Obj Material Swap | Self Con. (↓) | 43.20 | 18.63 | 23.79 | 17.28 | **16.18** | 29.48 | 24.76 |
| Scene Material Swap | Self Con. (↓) | 45.89 | **21.72** | 25.76 | 24.91 | 23.45 | 31.55 | 28.88 |
| Lighting | Self Con. (↓) | 30.28 | 12.90 | 16.66 | 14.18 | **12.60** | 22.90 | 18.25 |
| Obj Resizing | Self Con. (↓) | 21.58 | 7.41 | 12.32 | 8.84 | **7.27** | 17.93 | 12.56 |
| Average | Self Con. (↓) | 38.84 | **18.29** | 23.74 | 21.15 | 18.41 | 28.38 | 24.80 |

Table 18: Self-consistency of the $\delta_{0.125}$ metric across depth estimation models and perturbations. The scale of each prediction was aligned for evaluation. Best results are highlighted in bold.

| Perturbation Type | Metric | ZoeD | UniDV2 | Metric3DV2 | DepthPro | MoGe | VGGT | Avg |
|---|---|---|---|---|---|---|---|---|
| Cam Dolly Zoom | Avg. Err. (↓) | 38.24 | 47.84 | 48.51 | **24.60** | 36.09 | 40.81 | 39.35 |
| | Acc. (In)Stab. (↓) | 17.44 | 14.20 | 20.26 | **10.14** | 10.50 | 14.06 | 14.43 |
| Cam Roll | Avg. Err. (↓) | 44.74 | 43.00 | 45.01 | **37.64** | 40.79 | 54.63 | 44.30 |
| | Acc. (In)Stab. (↓) | 17.29 | 16.84 | 15.57 | **13.91** | 14.32 | 20.22 | 16.36 |
| Cam Pan/Tilt | Avg. Err. (↓) | 42.42 | 36.88 | 36.51 | **23.33** | 36.46 | 41.41 | 36.17 |
| | Acc. (In)Stab. (↓) | 16.18 | 12.53 | 13.35 | **6.90** | 9.07 | 12.01 | 11.67 |
| Obj Material Swap | Avg. Err. (↓) | 41.83 | 45.22 | 36.65 | **21.63** | 33.61 | 47.00 | 37.66 |
| | Acc. (In)Stab. (↓) | 17.91 | 16.51 | 10.61 | **7.23** | 12.53 | 26.34 | 15.19 |
| Scene Material Swap | Avg. Err. (↓) | 46.94 | 34.04 | 27.59 | **24.69** | 35.94 | 45.18 | 35.73 |
| | Acc. (In)Stab. (↓) | 19.41 | 16.02 | **9.94** | 12.24 | 18.09 | 29.29 | 17.50 |
| Lighting | Avg. Err. (↓) | 41.17 | 39.30 | 36.48 | **21.85** | 34.39 | 40.74 | 35.66 |
| | Acc. (In)Stab. (↓) | 12.23 | 12.16 | 11.56 | **7.83** | 9.62 | 14.73 | 11.35 |
| Obj Rotation | Avg. Err. (↓) | 38.67 | 44.99 | 46.17 | **24.08** | 35.10 | 38.92 | 37.99 |
| | Acc. (In)Stab. (↓) | 11.49 | 8.49 | 11.04 | **6.68** | 8.91 | 10.64 | 9.54 |
| Obj Resizing | Avg. Err. (↓) | 39.00 | 47.76 | 42.49 | **24.02** | 38.18 | 41.07 | 38.75 |
| | Acc. (In)Stab. (↓) | 9.34 | 9.64 | 11.34 | **5.91** | 8.03 | 11.66 | 9.32 |
| Obj Translation | Avg. Err. (↓) | 35.90 | 51.99 | 55.82 | **26.18** | 37.20 | 42.61 | 41.62 |
| | Acc. (In)Stab. (↓) | 11.83 | 16.19 | 23.69 | **7.23** | 9.95 | 12.70 | 13.60 |
| Obj Occlusion | Avg. Err. (↓) | 31.92 | 31.77 | 31.30 | **21.42** | 24.80 | 30.55 | 28.63 |
| | Acc. (In)Stab. (↓) | 11.41 | 12.23 | 11.18 | **6.39** | 11.07 | 12.62 | 10.82 |
| Non-Rigid Obj Deformation | Avg. Err. (↓) | 35.98 | 43.52 | 46.43 | **23.13** | 32.52 | 37.09 | 36.45 |
| | Acc. (In)Stab. (↓) | 14.27 | 12.73 | 14.87 | **6.85** | 9.02 | 11.11 | 11.48 |
| Average | Avg. Err. (↓) | 39.71 | 42.39 | 41.18 | **24.78** | 35.01 | 41.82 | 37.48 |
| | Acc. (In)Stab. (↓) | 14.44 | 13.41 | 13.95 | **8.30** | 11.01 | 15.94 | 12.84 |

Table 19: Performance of depth estimation models across different perturbations, measured in the error and stability of the RMSE metric (in cm). The scale of each prediction was aligned for evaluation. Best results are highlighted in bold.

| Perturbation Type | Metric | ZoeD | UniDV2 | Metric3DV2 | DepthPro | MoGe | VGGT | Avg |
|---|---|---|---|---|---|---|---|---|
| Cam Roll | Self Con. (↓) | 7.07 | 5.03 | 6.04 | **4.67** | 4.89 | 8.08 | 5.96 |
| Obj Material Swap | Self Con. (↓) | 5.50 | 5.50 | 4.59 | **2.29** | 3.71 | 5.37 | 4.49 |
| Scene Material Swap | Self Con. (↓) | 5.52 | 4.70 | 4.11 | **3.07** | 4.50 | 6.25 | 4.69 |
| Lighting | Self Con. (↓) | 3.58 | 4.41 | 4.11 | **2.07** | 3.12 | 4.45 | 3.62 |
| Obj Resizing | Self Con. (↓) | 2.86 | 3.40 | 3.69 | **1.71** | 2.54 | 3.81 | 3.00 |
| Average | Self Con. (↓) | 4.90 | 4.61 | 4.51 | **2.76** | 3.75 | 5.59 | 4.35 |

Table 20: Self-consistency of the RMSE metric (in cm) across depth estimation models and perturbations. The scale of each prediction was aligned for evaluation. Best results are highlighted in bold.

# F Broader Impacts

Robust depth estimation is a critical component in robots, self-driving cars, and other autonomous AI systems that interact with humans and objects in the world. Premature deployment of systems which are not suitably robust could pose dangers to humans as well as public or private property. Our evaluation reveals some of the manners in which current systems are not sufficiently robust. In this sense, we hope to further the development of safe and robust models. However, we do not intend for our evaluation to act as a sufficient guarantor of robustness, and individuals deploying systems in the real-world must consider many additional factors.

