# OpenReview forum: "Evaluating Robustness of Monocular Depth Estimation with Procedural Scene Perturbations"
_NeurIPS.cc/2025/Conference — NeurIPS 2025 poster_

### Official Review · Reviewer_j7wK · 2025-06-10

**Clarity:** 3
**Significance:** 3
**Originality:** 3
**Rating:** 4
**Confidence:** 3

**Summary:**

This paper proposes to evaluate a crucial, though not exhaustive, aspect of depth estimation models: robustness. The evaluation is conducted by introducing procedural scene perturbations to generate diverse variations of the original scenes. The authors perform an extensive assessment of recent state-of-the-art models and identify which types of perturbations each model is more robust to, compared to others.

**Questions:**

- I would like the authors to clarify why perturbations on synthetic datasets are considered representative of real-world generalization.
- It would be valuable to see results for combinations of different perturbations, even if only a few combinations are tested.
- I am interested in seeing how VGGT[1] performs under these perturbations. Since it is primarily trained with multi-input images, it may exhibit greater robustness to camera-related perturbations.

[1] Wang J, Chen M, Karaev N, et al. Vggt: Visual geometry grounded transformer[C]//Proceedings of the Computer Vision and Pattern Recognition Conference. 2025: 5294-5306.

**Ethical Concerns:**

["NO or VERY MINOR ethics concerns only"]

**Final Justification:**

I thank the authors for their rebuttal. My assessment remains a boarderline accept.

The paper makes a clear and valuable contribution by introducing what appears to be the first systematic benchmark for the robustness of monocular depth estimation models. By diagnosing specific weaknesses, it offers valuable insights that can guide the design of more robust future models. For these contributions, the work warrants acceptance.

However, my score is limited because the evaluation—which focuses on individual synthetic perturbations—does not fully support the broader claims about performance in complex, real-world robustness where multiple factors interact. The work is therefore more accurately positioned as a crucial diagnostic probe.

While this is an important and novel contribution, the need for this significant reframing is the primary reason for my score.

**Limitations:**

yes

**Quality:**

3

**Strengths And Weaknesses:**

**Strengths:**

- The authors introduce an important, though not comprehensive, evaluation perspective for existing monocular depth estimation (MDE) models: robustness, which is a critical factor for practical deployment.
- The study conducts extensive evaluations on recent state-of-the-art methods.
- The paper reports some findings(L73-79) which highlight areas that could guide future improvements.

**Weaknesses:**

- My main concern is the data source: since the perturbations are applied primarily to a synthetic dataset (Infinigen), evaluating **robustness** solely on **synthetic** data may not provide sufficiently persuasive results.
- The paper evaluates the effect of one perturbation at a time, whereas real-world out-of-distribution (OOD) images often involve multiple factors combined (e.g.,transparent object + dim lighting + out-of-focus blur).
- Table 3 is confusing; it appears to report self-(in)consistency, yet the metric is again listed as Avg.Err, which is unclear.

---

> ### Author Rebuttal · Authors · 2025-07-31
>
> Thanks for your constructive feedback, and we are glad that you found the importance of introducing robustness to monocular depth estimation (MDE) assessment and that our findings could guide future improvements. We have addressed the individual concerns below.
>
> > Concern 1 (and question 1): Results on synthetic data may not generalize to the real world.
>
> While we acknowledge concerns about the domain gap, we argue that synthetic evaluation provides invaluable insights that complement measurements with real-world data. Real-world depth faces fundamental limitations that are addressed by synthetic data: (1) challenges obtaining dense ground truth depth in the presence of specular surfaces, transparent objects, and distant objects (2) limited scene variety (e.g. no underwater scenes) (3) prohibitive costs when creating controlled and precise perturbations at a large scale. In the context of our experiments, synthetic data can effectively reveal flaws in models’ performance by enabling precise comparisons for many scenes and perturbations. Notably, this includes perturbations such as object deformation which cannot be easily replicated in the real world.
>
> In addition, our synthetic images are legible to humans in terms of their 3D structure. The fact that models struggle on them reveals potential misalignment of models with human perception, which can help guide model development.
>
> To address this question about generalizability we performed preliminary real-world experiments using a mug with a known 3D structure. We then captured 15-20 images per variation and manually annotated the pose and object mask to infer a precise depth map for the object. We present the results below. Since this is a limited experiment in the context of objects on a table top (similar to the domain of robotic arms), one should not expect the results to perfectly match our much more diverse synthetic evaluation.
>
> **Error**
> | Test Scenario | DepthAnything | DepthAnythingV2 | DepthPro | Marigold | Metric3DV2 | MiDaS | MoGe | UniDepth | ZoeDepth | Average |
> |---------------|---------------|-----------------|----------|----------|------------|-------|------|----------|----------|---------|
> | Cam Pan/Tilt | 2.31 | 1.59 | 1.66 | 2.31 | 1.82 | 2.73 | 1.81 | 1.04 | 2.87 | 2.01 |
> | Camera Roll | 1.92 | 1.23 | 1.43 | 2.13 | 1.16 | 1.96 | 1.80 | 1.16 | 2.05 | 1.65 |
> | Lighting | 2.33 | 1.62 | 1.66 | 2.38 | 1.62 | 2.63 | 1.81 | 1.22 | 2.60 | 1.98 |
> | Occlusion | 2.62 | 1.73 | 1.92 | 2.70 | 2.11 | 2.89 | 2.15 | 1.50 | 2.98 | 2.29 |
> | Rotate | 2.28 | 1.73 | 1.76 | 2.75 | 1.91 | 2.65 | 2.15 | 1.52 | 2.76 | 2.17 |
> | Average | 2.29 | 1.58 | 1.69 | 2.45 | 1.72 | 2.57 | 1.94 | 1.29 | 2.65 | 2.02 |
>
> **Stability**
> | Category | DepthAnything | DepthAnythingV2 | DepthPro | Marigold | Metric3DV2 | MiDaS | MoGe | UniDepth | ZoeDepth | Average |
> |----------|---------------|-----------------|----------|----------|------------|-------|------|----------|----------|---------|
> | cam_pan_tilt | 0.1846 | 0.3564 | 0.4183 | 0.3168 | 0.3666 | 0.2016 | 0.3341 | 0.2513 | 0.1217 | 0.2835 |
> | camera_roll | 0.3987 | 0.3414 | 0.3879 | 0.4799 | 0.2666 | 0.3828 | 0.4225 | 0.3122 | 0.5408 | 0.3925 |
> | lighting | 0.2499 | 0.2627 | 0.2516 | 0.4311 | 0.3376 | 0.2002 | 0.3731 | 0.4483 | 0.2947 | 0.3166 |
> | occlusion | 0.4359 | 0.4676 | 0.2591 | 0.4003 | 0.5227 | 0.5714 | 0.3788 | 0.4672 | 0.5548 | 0.4509 |
> | rotate | 0.5684 | 0.4389 | 0.4624 | 0.6950 | 0.3235 | 0.6661 | 0.5379 | 0.3700 | 0.7202 | 0.5314 |
> | avg | 0.3675 | 0.3734 | 0.3559 | 0.4646 | 0.3634 | 0.4044 | 0.4093 | 0.3698 | 0.4464 | 0.3950 |
>
>
> We find similar though not identical trends in terms of models’ average performance, with UniDepth, DepthAnythingV2, and DepthPro remaining as 3 of the top 4 models by average error. We find that occlusion remains a difficult task and many models remain fairly robust to changes in lighting. We will include a more thorough analysis in our final version.
>
>
> > Concern 2 (and question 2): Only one perturbation is evaluated at the time, while real-world images often involve multiple factors combined. It would be valuable to see the results for combinations of perturbations
>
> We believe testing individual perturbations is sufficient for evaluating robustness. Real-world scenes are not simply “clean” scenes with perturbations added; they naturally contain a mix of challenging factors. Since our base scenes already exhibit substantial diversity, applying perturbations individually already captures a wide range of realistic conditions.
>
> Some of the perturbations (in-distribution perturbations) already exist in the base scenes, e.g., diverse lighting conditions, materials, and camera tilt angles; some of them (out-of-distribution perturbations) do not, e.g., camera roll angles are typically standardized in the base scene. For those in-distribution perturbations, we expect that combining perturbations will have at most a small impact on the average error, while the instability may increase due to the greater difference between the perturbed scene and the original. When combining out-of-distribution perturbations, we would expect the average error to increase, though the exact effect is unclear.
>
> We present preliminary results for combinations of perturbations. Specifically, we generate the variations Camera Pan/Tilt + Translate Object, Camera Roll + Object Material Swap, and Translate Object + Scene Material Swap for chairs and desks. The following tables show the results averaged over all models for the original perturbations and the combined perturbations for the objects chairs and desks.
>
>
> |                   | Camera Roll | Object Translation | Object Material Swap | Camera Pan/Tilt | Scene Material Swap |
> |-------------------|-------------|---------------------|------------------------|------------------|----------------------|
> | Error     | 2.96        | 2.46 | 2.44   | 2.37     | 2.53      |
> | Acc. (In)Stability | 0.57        | 0.36  | 0.35   | 0.41   | 0.47     |
>
> |                                   | Camera Roll + Object Material Swap | Cam Pan/Tilt + Translate Object | Translate Object + Scene Material Swap |
> |----------------------------------|------------------------------------|----------------------------------|----------------------------------------|
> | Error       | 3.20                               | 2.46                             | 2.58                                   |
> | Acc. (In)Stability         | 0.56                               | 0.51                             | 0.50                                   |
>
>
> This confirms our expectations about the interactions between variations. For variations that introduce out-of-distribution changes, such as camera roll + material swap, the average error compounds when perturbing the scene with both variations. For object translation, which produces in-distribution perturbed scenes, the average error is close to the maximum of the two original errors. We do find for Cam Pan/Tilt + Translate Object that the instability increases, which is sensible since the difference from the original image is now larger.
>
> > Concern 3: Confusing labeling of Table 3
>
> We will update this to read Self-(in)consistency.
>
>
> > Question 3: I’m interested to see results of VGGT [1] – does multi-image training lead to greater robustness to camera perturbations?
>
> We present results for VGGT below.
> | | VGGT Error | VGGT Stab | VGGT Self-Consistency |
> |---|---|---|---|
> | Dolly Zoom | 2.19 | 0.63 | |
> | Camera Roll | 2.08 | 0.34 | 3.55 |
> | Camera Pan + Tilt | 1.88 | 0.31 | |
> | Object Material Swap | 1.92 | 0.31 | 2.40 |
> | Scene Material Swap | 1.87 | 0.37 | 2.63 |
> | Lighting | 1.83 | 0.18 | 1.90 |
> | Object Rotation | 1.92 | 0.30 | |
> | Object Resizing | 1.95 | 0.19 | 1.79 |
> | Object Translation | 1.93 | 0.29 | |
> | Object Occlusion | 2.01 | 0.36 | |
> | Object Deformation | 1.87 | 0.39 | |
> | Average | 2.15 | 0.33 | 2.45 |
>
> VGGT is not competitive with top models. For average of all three metrics over perturbation types, VGGT is worse than Depth Pro and UniDepthV2) The poor average error might be due to VGGT outputs depth maps at a low resolution (294x518). To perform evaluation, we upsample the depth maps using nearest neighbor interpolation.
> Regarding metrics that evaluate robustness to camera-related perturbations (Accuracy (in)stability and Self-(in)consistency on Cam Dolly Zoom, Cam Roll, Cam Pan/Tilt), it is also not the best. This may suggest that multi-image training may not directly lead to greater robustness to camera-related perturbations. (Note that low operating resolution does not affect robustness much, as predictions for base is also made under low operating resolution).
>
> [1] Wang J, Chen M, Karaev N, et al. Vggt: Visual geometry grounded transformer[C]//Proceedings of the Computer Vision and Pattern Recognition Conference. 2025: 5294-5306.

---

> > ### Comment · Reviewer_j7wK · 2025-08-02
> > **Rebuttal Reply**
> >
> > I thank the authors for their clear and insightful rebuttal. While I agree that their synthetic perturbations are valuable for assessing performance, my primary reservation is that the paper, in its current form, overclaims its contribution.
> >
> > Therefore, I would strongly encourage the authors to more precisely position their work in the next version. The paper's contribution would be most accurately framed as the development of a synthetic benchmark to serve as a diagnostic probe for understanding the robustness of monocular depth estimation models.
> >
> > This reframing is essential for accurately representing the work's scope. While this is a valuable contribution, my overall assessment remains unchanged. I will maintain my original score.

---

### Official Review · Reviewer_rtFi · 2025-06-27

**Clarity:** 2
**Significance:** 3
**Originality:** 2
**Rating:** 4
**Confidence:** 3

**Summary:**

To evaluate the robustness of monocular depth models, the paper introduces a synthetic dataset created with the procedural generator Infinigen. The dataset includes 5 objects, 38 distinct scenes and 12 controlled perturbations. The paper performs a comprehensive evaluation of the robustness of 9 state-of-the-art monocular depth models.

**Questions:**

- It would be good to explain the motivation for the design of some perturbations. For example, why the 2D size of objects have to be roughly maintained for perturbation e.g. Camera Dolly Zoom? Is it to keep the saliency and the visibility?
- In L207-208, it is mentioned that *when evaluating on an object of interest, we compute alignment using only the depth values for the object*. Would it lead to metrics that capture only local robustness/consistency and potentially missing the global inconsistency among the predictions? What if the alignment is computed on the entire scene and then the error metrics are computed on the object area?

**Ethical Concerns:**

["NO or VERY MINOR ethics concerns only"]

**Final Justification:**

I was torn between a score of 3 and 4 for this paper. My initial concern was the lack of full scene evaluation and whether the objects used for evaluation were salient enough. The authors provided additional results that helped alleviate these concerns to some extent. Ultimately, I leaned toward a score of 4, assuming the additional results will be included in the final version.

**Paper Formatting Concerns:**

No formatting concern

**Quality:**

3

**Strengths And Weaknesses:**

**Strengths**
- Understanding the robustness of models to various perturbations is crucial, and the new dataset could offer a valuable tool for the research community.
- The paper is well-written and easy to follow. Table 1 and Figure 1 provide a good example and summary of the involved perturbation types.

**Weaknesses**
- In L119, the paper states, *“We ensure the object of interest is salient in the scene”*. However, It is unclear how this is guaranteed. Furthermore, according to the Kaggle link provided in the appendix, for objects e.g.fish, in the scene 3674e176 , the target fish object only occupies a very small portion of the image, which raises questions about the claim of salience.
- In Section 3.1, the perturbation type Camera Pedestal/Truck is mentioned, but it is not included among the 12 scene perturbations listed in Table 1 and Figure 1. If it is not part of the dataset, it should be removed from the text to avoid confusion.
- In Table 3, according to the title the reported metrics are self-(in)consistency, while it shows as Avg.Err in the table which is the same as the metric name in Table 2, and it could be confusing.
- There are no evaluation results based on the full scene, only object-based results are reported. Including full-scene results would provide a more comprehensive understanding of the models’ performance.

---

> ### Author Rebuttal · Authors · 2025-07-31
>
> Thanks for your constructive feedback, and we are glad that you found our paper well-written and that our benchmark is valuable in understanding the robustness of models to various perturbations. We have addressed the individual concerns below.
>
> > Concern 1: It is unclear how we guarantee that the object of interest is sufficiently salient in the scene given that the fish in scene 3674e176 is fairly small.
>
> The particular fish mentioned is the third smallest of all objects, has an object mask with 3970 pixels, and has a bounding box size of 7685 pixels; it would be classified as medium size by COCO’s object categories. To ensure that objects are sufficiently salient we generate scenes with Infinigen and then manually select specific objects and scenes. We sought to encompass a reasonable distribution of object placement – we didn’t want all objects to be large and perfectly centered in the frame, but we also didn't want them to be occluded by other objects. To get a better sense of the distribution, the median object has a bounding box size of 92308 px (~= $304^2$ px). Using COCO’s small/medium/large object categories (appropriately scaled to our image resolution) we find we have 0 small objects, 8 medium objects, and 32 large objects.
>
> > Concern 2: Reference to camera pedestal/truck in section 3.1
>
> Thanks for pointing this out. This variation is not present in the dataset and we will correct this section to remove the reference.
>
> > Concern 3: Labeling of Table 3 is confusing/incorrect.
>
> We will update this to read Self-(in)consistency.
>
> > Concern 4: There are no evaluation results on the full scene
>
> In figure 2 (L212) we display the average AbsRel error on the full scene. We elected to focus on object-level comparisons since these are much more precise and avoid the ambiguity present in full scene MDE. We still do agree that there are interesting insights to be gained from more closely analyzing robustness on the entire scene. We present a summary of the results below.
>
> | Metric | camera_dolly_zoom | camera_roll | camera_pan_tilt | object_material_swap | scene_material_swap | lighting | object_rotation | object_resizing | object_translation | object_occlusion | object_deformation | avg |
> |--------|-------------------|-------------|-----------------|----------------------|---------------------|----------|-----------------|-----------------|--------------------|-----------------|--------------------|---------|
> | Avg Error | 8.13 | 14.39 | 8.64 | 7.91 | 10.26 | 7.98 | 10.05 | 8.32 | 7.89 | 8.56 | 7.91 | 9.09 |
> | Accuracy (In)stability | 4.57 | 6.23 | 4.97 | 0.35 | 2.01 | 1.20 | 1.31 | 2.87 | 0.49 | 0.53 | 0.35 | 2.26 |
> | Self-(In)consistency | - | - | - | - | 9.10 | 7.58 | - | - | - | - | - | 8.34 |
>
>
> | Metric | MiDaS | DepthAnything | DepthAnythingV2 | ZoeDepth | UniDepth | Metric3DV2 | Marigold | DepthPro | MoGe | Average |
> |--------|-------|---------------|-----------------|----------|----------|------------|----------|----------|------|-------------|
> | Avg Error | 9.76 | 8.73 | 8.12 | 12.52 | 7.43 | 8.21 | 12.25 | 8.25 | 6.58 | 9.09 |
> | Accuracy (In)stability | 2.61 | 2.19 | 1.87 | 2.71 | 1.61 | 1.77 | 3.04 | 2.11 | 2.46 | 2.26 |
> | Self-(In)consistency | - | - | - | 5.20 | 12.29 | 12.52 | - | 5.51 | 6.16 | 8.34 |
>
> Among the variations, the instability largely aligns with the expectations based on the magnitude of the pixel-level change in the scene. For example, object material swap only affects the relatively small area of the object and so has very low instability. Notably, lighting also has low instability despite impacting larger portions of the scene. Among the different models, we observe that MoGe achieves the best performance by average error while UniDepth has the best accuracy (in)stability. We will include complete tables and more thorough analysis in the appendix.
>
> > Question 1: What is the motivation for some perturbations? Why is the 2D size of objects roughly maintained? Is it to keep the saliency and the visibility?
>
> For each perturbation, we sought to make minimal alterations to the depth map of the object so that the task complexity is unchanged and performance on the perturbed scenes can be directly compared. Saliency and visibility are important components of this since there may be less visual information if the object is smaller. However zooming in on the object while maintaining its apparent size will preserve geometric information. Similarly in the case of object translation and rotation we only select perturbations in a “small neighborhood” so that the depth map is not dramatically changed by revealing another side of the object.
>
> > Question 2: Would computing alignment using only the depth values for the object lead to metrics that capture only local robustness/consistency and potentially missing the global inconsistency? What if the alignment is computed on the entire scene and then the error metrics are computed on the object area?
>
> Yes, alignment using only the depth values for the object misses global inconsistencies. As discussed above, we agree that global consistency is important and can be captured with metrics on the full scene.
>
> However, we do not find it reasonable to compute the alignment on the entire scene and only evaluate metrics on the object area. We design our perturbations such that the complexity of the task of predicting the object geometry is unchanged, but the same does not hold for predicting the background geometry. For example, a camera pan/tilt or lighting variation may alter the background such that its depth becomes ambiguous. Then, computing alignment on the background image could introduce distortions which even a robust model would be unable to avoid. Moreover, it would be difficult to interpret the magnitude of the error and its implications for assessing the model.

---

> > ### Comment · Reviewer_rtFi · 2025-08-04
> >
> > Thanks for the response and for providing the additional results on the full scene.
> >
> > I agree alignment with the object area only makes more sense given the proposed perturbation, and the full scene results could capture the global inconsistencies.
> >
> > > The particular fish mentioned is the third smallest of all objects, has an object mask with 3970 pixels, and has a bounding box size of 7685 pixels; it would be classified as medium size by COCO’s object categories.
> >
> > The resolution of the proposed dataset is 1280*720, so an object with 3970 pixels is only 0.4% of the entire image. Considering baseline methods may require downsample resizing for inference, I remain concerned that such examples may not be sufficiently salient for a meaningful evaluation.

---

> > > ### Author Response · Authors · 2025-08-06
> > >
> > > Thank you for your feedback and for further clarification of your concern.
> > >
> > > Though this particular fish is small, we argue that it is still fair to evaluate models on objects of this size. For methods that operate at a high resolution, 3970 pixels should be sufficient to understand the shape of the object. In fact, this is an advantage in contexts such as robotics where we might want to understand the shape of small objects being grasped.
> > >
> > > That said, it is a fair concern that current baselines may perform downsampling and may not be optimized for small objects, and our evaluation may not reflect their true capability. Here we show that this is not the case and that excluding small objects gives qualitatively similar results and rankings.
> > >
> > > Table 1 displays the operating resolution of the models we test. 6 out of the 9 models retain at least 50% of the pixel count of the original 1280 x 720 image. However, MiDaS, Marigold, and ZoeDepth each downsample the image by larger factors.
> > >
> > > Table 2 displays the results of aggregating while excluding all objects which are not in the COCO-large category. This removes 8 out of 40 objects, and all remaining objects have a bounding box enclosing at least 3% of the image (greater than a $1/6H\times1/6W$ rectangle).
> > >
> > > This change has minimal impact on the rankings of the models. Meanwhile, it leads to an overall increase in error and instability metrics. This suggests  that small objects are in fact easier than the average object and are not a dominant source of difficulty.
> > >
> > > If it were the case that image downsampling made the small objects difficult for certain models, we would expect the performance of low-resolution models (ZoeDepth, MiDaS, Marigold) to grow closer to the performance of high-resolution models (UniDepthV2, Metric3DV2, DepthPro) after the small objects are removed. However, this does not occur. The gap in the average error between high resolution models and low resolution models increases from 0.96 to 1.05 after removing the small objects. The gap in the average accuracy instability increases from 0.12 to 0.13. Since these gaps do not grow smaller, this suggests that the presence of small objects is not a determining factor in our evaluation.
> > >
> > > However, we do agree that object size is an important factor to consider. In the next version  we will include additional analysis breaking down the impact of object size on model performance.
> > >
> > > **Table 1:** Model Internal Resolution
> > >
> > > | Metric | ZoeDepth | MiDaS | Marigold | DepthAnythingV2 | DepthAnything | MoGe | UniDepthV2 | Metric3DV2 | DepthPro |
> > > |--------|----------|--------|----------|-----------------|---------------|------|----------|------------|----------|
> > > | Internal Resolution | 512 x 384 [1] | 512 x 512 [2,3] | 640 x 480 [4] | 920 x 518 [5] | 920 x 518 [6] | 700 x 700 [7] | 774 x 774 [7] | 1064 x 616 [7] | 1536 x 1536 [7] |
> > > | % Original Pixels | 21% | 28% | 33% | 52% | 52% | 53% | 65% | 71% | 256% |
> > >
> > > **Table 2:** Model Performance Comparison with large objects
> > >
> > > | Metric | ZoeDepth | MiDaS | Marigold | DepthAnythingV2 | DepthAnything | MoGe | UniDepthV2 | Metric3DV2 | DepthPro |
> > > |--------|----------|--------|----------|-----------------|---------------|------|----------|------------|----------|
> > > | Original Avg. Error | 2.64 | 2.45 | 2.35 | 1.77 | 1.99 | 1.52 | 1.47 | 1.78 | 1.30 |
> > > | Large Objects Avg. Error | 2.78 | 2.58 | 2.52 | 1.81 | 2.05 | 1.58 | 1.51 | 1.85 | 1.35 |
> > > | % Degradation | 5.32% | 5.45% | 7.27% | 2.47% | 3.25% | 3.97% | 3.23% | 3.52% | 4.09% |
> > > | Original Acc. (In)Stability | 0.41 | 0.40 | 0.40 | 0.29 | 0.32 | 0.31 | 0.27 | 0.29 | 0.30 |
> > > | Large Objects Acc. (In)Stability | 0.44 | 0.43 | 0.43 | 0.31 | 0.34 | 0.33 | 0.29 | 0.29 | 0.33 |
> > > | % Degradation | 6.09% | 7.62% | 8.09% | 4.99% | 7.03% | 5.15% | 8.24% | 1.75% | 9.07% |
> > > | % Original Pixels | 21% | 28% | 33% | 52% | 52% | 53% | 65% | 71% | 256% |
> > >
> > > [1] Bhat, Shariq Farooq, et al. "Zoedepth: Zero-shot transfer by combining relative and metric depth." arXiv preprint arXiv:2302.12288 (2023).
> > >
> > > [2] Birkl, Reiner, et al. “MiDaS.” https://github.com/isl-org/MiDaS
> > >
> > > [3] Birkl, Reiner, Diana Wofk, and Matthias Müller. "Midas v3. 1--a model zoo for robust monocular relative depth estimation." arXiv preprint arXiv:2307.14460 (2023).
> > >
> > > [4] Ke, Bingxin, et al. "Repurposing diffusion-based image generators for monocular depth estimation." Proceedings of the IEEE/CVF conference on computer vision and pattern recognition. 2024.
> > >
> > > [5] Yang, Lihe, et al. "Depth anything v2." Advances in Neural Information Processing Systems 37 (2024): 21875-21911.
> > >
> > > [6] Yang, Lihe, et al. "Depth anything: Unleashing the power of large-scale unlabeled data." Proceedings of the IEEE/CVF conference on computer vision and pattern recognition. 2024.
> > >
> > > [7] Wang, Ruicheng, et al. "MoGe-2: Accurate Monocular Geometry with Metric Scale and Sharp Details." arXiv preprint arXiv:2507.02546 (2025).

---

> > > > ### Comment · Reviewer_rtFi · 2025-08-07
> > > >
> > > > Thank you for the thorough respond and the additional results.
> > > > The results in Table 2 suggests that excluding small objects has minimal effect, so I think my main concern actually get addressed and I'm willing to increase my rating to 4.

---

### Official Review · Reviewer_c7NB · 2025-07-02

**Clarity:** 3
**Significance:** 2
**Originality:** 3
**Rating:** 4
**Confidence:** 4

**Summary:**

This paper evaluates the robustness of state-of-the-art monocular depth estimation models (e.g., MiDaS, DepthAnything, ZoeDepth) to procedural scene perturbations, including changes in camera parameters, lighting, materials, and object geometry. Using synthetic 3D scenes generated via Infinigen, the authors systematically test model performance under controlled perturbations, measuring both accuracy stability (consistency against ground truth) and self-consistency (prediction stability across perturbations). Key findings include:
Models are robust to lighting changes but struggle with camera perturbations (e.g., dolly zoom, roll) and material swaps.
Occlusions and non-rigid deformations are particularly challenging, revealing limitations in geometric reasoning.
DepthPro and UniDepthV2 emerge as top performers, though trade-offs exist between accuracy and robustness.
The study highlights gaps in standard benchmarks (e.g., KITTI, NYU-D) by focusing on robustness rather than just accuracy, providing insights for real-world applications like robotics.

**Questions:**

Questions

1. Why were certain perturbation types (e.g., camera dolly zoom) prioritized over others (e.g., motion blur or weather effects)? How were perturbation parameters (e.g., occlusion percentage) determined?

2. For perturbations that alter ground truth depth (e.g., object translation), how do you justify the "small neighborhood" constraint? Could larger perturbations reveal additional failure modes?

3. Self-consistency is undefined for affine-invariant models (e.g., MiDaS). Did you explore alternative metrics (e.g., scale-invariant error) to include these models in consistency analysis?

4. AbsRel error emphasizes large depth errors but may miss local inconsistencies. Did you consider edge-aware metrics (e.g., gradient-based errors) to evaluate geometric fidelity?

5.Infinigen scenes lack real-world noise (e.g., sensor artifacts). How would results change if tested on real-world datasets with paired perturbations (e.g., Replica-CE)?
The object categories (chairs, fish, etc.) are limited. Would results generalize to complex urban scenes (e.g., crowded streets)?

6.The paper identifies camera perturbations as highly challenging. How should robotics/AR systems mitigate this in practice (e.g., sensor fusion, temporal smoothing)?

7. Will the perturbation code support custom scenes (e.g., Unity/Unreal Engine assets) beyond Infinigen?

8. Could this framework evaluate video-based depth estimation (e.g., robustness to temporal inconsistencies)?

**Ethical Concerns:**

["NO or VERY MINOR ethics concerns only"]

**Final Justification:**

The author's response solved my problem and I improved the score to  boarderline accept.

**Limitations:**

yes

**Paper Formatting Concerns:**

no formatting issue

**Quality:**

2

**Strengths And Weaknesses:**

Strengths
1. This work introduces a procedurally generated dataset with controlled perturbations, enabling systematic robustness testing beyond standard benchmarks.

2. Tests 12 perturbation types (e.g., camera intrinsics, material swaps, occlusions) across 9 models, revealing understudied failure modes (e.g., sensitivity to focal length changes).
Includes object-level evaluation (not just full-scene metrics) for finer-grained insights.

3. This work identifies camera perturbations as a critical weakness, informing deployment in dynamic environments (e.g., autonomous vehicles). Highlights material invariance as an open challenge, suggesting directions for future work (e.g., physics-aware training).

4. Plans to release open-source code for benchmarking new models, fostering community engagement.


Weaknesses
1. The work relies solely on synthetic data (Infinigen), which may not fully capture real-world noise (e.g., sensor artifacts, texture variations).
Domain gap could affect generalizability, despite Infinigen’s photorealism.

2.Experiment tests perturbations individually, while real-world scenarios often involve combined perturbations (e.g., lighting + occlusion).

3. The coverage of urban/driving scenes is limited, which are critical for applications like autonomous navigation.

4. Model Scope excludes recent foundation models (e.g., those leveraging diffusion or multimodal pretraining), which may exhibit different robustness profiles.

5. The is no analysis of computational efficiency vs. robustness trade-offs (e.g., latency for real-time systems).

---

> ### Author Rebuttal · Authors · 2025-07-31
>
> > Concern 1: Results on synthetic data may not generalize to the real world
>
> While we acknowledge concerns about the domain gap, we argue that synthetic evaluation provides invaluable insights that complement measurements with real-world data. Real-world depth faces fundamental limitations that are addressed by synthetic data: (1) challenges obtaining dense ground truth depth in the presence of specular surfaces, transparent objects, and distant objects (2) limited scene variety (e.g. no underwater scenes) (3) prohibitive costs when creating controlled and precise perturbations at a large scale. In the context of our experiments, synthetic data can effectively reveal flaws in models’ performance by enabling precise comparisons for many scenes and perturbations. Notably, this includes perturbations such as object deformation which cannot be easily replicated in the real world.
>
> To address this question about generalizability we performed preliminary real-world experiments using a mug with a known 3D structure. We captured 15-20 images per variation and manually annotated the object’s pose in order to compute a precise depth map. Since this experiment is limited to a single object on a table, one should not expect the results to perfectly match our much more diverse synthetic evaluation. We display selected models below and will include the full results in the paper.
>
>
> **Error**
> | Scenario | DepthAnythingV2 | DepthPro | Metric3DV2 | MoGe | UniDepth | Average |
> |---|---|---|---|---|---|---|
> | Pan/Tilt | 1.59 | 1.66 | 1.82 | 1.81 | 1.04 | 2.01 |
> | Roll | 1.23 | 1.43 | 1.16 | 1.80 | 1.16 | 1.65 |
> | Lighting | 1.62 | 1.66 | 1.62 | 1.81 | 1.22 | 1.98 |
> | Occlusion | 1.73 | 1.92 | 2.11 | 2.15 | 1.50 | 2.29 |
> | Rotate | 1.73 | 1.76 | 1.91 | 2.15 | 1.52 | 2.17 |
> | Average | 1.58 | 1.69 | 1.72 | 1.94 | 1.29 | 2.02 |
>
>
> **Stability**
> | Category | DepthAnythingV2 | DepthPro | Metric3DV2 | MoGe | UniDepth | Average |
> |---|---|---|---|---|---|---|
> | Pan/Tilt | 0.36 | 0.42 | 0.37 | 0.33 | 0.25 | 0.28 |
> | Roll | 0.34 | 0.39 | 0.27 | 0.42 | 0.31 | 0.39 |
> | Lighting | 0.26 | 0.25 | 0.34 | 0.37 | 0.45 | 0.32 |
> | Occlusion | 0.47 | 0.26 | 0.52 | 0.38 | 0.47 | 0.45 |
> | Rotate | 0.44 | 0.46 | 0.32 | 0.54 | 0.37 | 0.53 |
> | Average | 0.37 | 0.36 | 0.36 | 0.41 | 0.37 | 0.40 |
>
> We find similar though not identical trends in terms of models’ average performance, with UniDepth, DepthAnythingV2, and DepthPro remaining as 3 of the top 4 models by average error. We find that occlusion remains a difficult task and many models remain fairly robust to changes in lighting. We will include a more thorough analysis in our final version.
>
> > Concern 2: Real-world scenarios often involve combined perturbations.
>
> We believe testing individual perturbations is sufficient for evaluating robustness. Real-world scenes are not simply “clean” scenes with perturbations added; they naturally contain a mix of challenging factors. Since our base scenes already exhibit substantial diversity, applying perturbations individually already captures a wide range of realistic conditions.
>
> We present preliminary results for combinations of perturbations. Specifically, we generate the variations Camera Pan/Tilt + Translate Object, Camera Roll + Object Material Swap, and Translate Object + Scene Material Swap for chairs and desks. The following tables show the results averaged over all models for the original perturbations and the combined perturbations for the objects chairs and desks.
>
> | | Camera Roll | Object Translation | Object Material Swap | Camera Pan/Tilt | Scene Material Swap |
> |---|---|---|---|---|---|
> | Error | 2.96 | 2.46 | 2.44 | 2.37 | 2.53 |
> | Acc. (In)Stability | 0.57 | 0.36 | 0.35 | 0.41 | 0.47 |
>
> | | Camera Roll + Object Material Swap | Cam Pan/Tilt + Translate Object | Translate Object + Scene Material Swap |
> |---|---|---|---|
> | Error | 3.20 | 2.46 | 2.58 |
> | Acc. (In)Stability | 0.56 | 0.51 | 0.50 |
>
> These results align with our expectations. For variations that introduce out-of-distribution changes, such as camera roll + material swap, the average error compounds when perturbing the scene with both variations. For object translation, which produces in-distribution perturbed scenes, the average error is close to the maximum of the two original errors. We do find for Cam Pan/Tilt + Translate Object that the instability increases, which is sensible since the difference from the original image is now larger.
>
> > Concern 3 (+ question 5): Coverage of urban/driving scenes is limited
>
> We do not explicitly cover urban/driving scenes and their associated objects, though we do expect our results to generalize to new situations such as these. We evaluate the robustness of the geometry estimation of particular objects, with our selection of objects encompassing different geometric attributes – thin structure (chair, desk), planar shapes (desk, cabinet), irregular curved surfaces (fish, cactus). We expect that a model that can robustly predict these geometries would succeed in an urban driving environment, though we leave this to future work.
>
> > Concern 4: “Model Scope excludes recent foundation models”
>
> We respectfully disagree; our models are selected to encompass recent state-of-the-art advances in monocular depth estimation and geometry estimation. Of the 9 models we evaluate, 3 were presented at conferences or published on arxiv.org in 2025, and an additional 4 were presented or published in 2024. This includes the diffusion model Marigold and models which predict multiple modalities – Depth Pro, UniDepthV2, MoGe, and Metric3DV2.
>
> We additionally present results for Geowizard [Fu et al, 2024] and VGGT [Wang et al, 2025] (published March 14, 2025, concurrent with NeurIPS submission). Geowizard extends the Stable Diffusion [Rombach et al, 2022] model to predict depth + normal, while VGGT predicts a point cloud from multiple input images.
>
> We find that neither Geowizard nor VGGT display results competitive with the best monocular depth estimation models. We summarize the results below and will include full results in the final paper.
>
> | Test Scenario | VGGT Error | VGGT Acc. Stability | Geowizard Error | Geowizard Stability |
> |---|---|--|-----|---|
> | **Average** | **2.15** | **0.33** | **2.60** | **0.44** |
>
>
> > Concern 5: “There is no analysis of computational efficiency vs. robustness tradeoffs”
>
> In figure 8 (L277) and section 4.3 (L279-286) we analyze the robustness of various models compared to their FLOPs.
>
> We additionally measure the runtime for four of the top-performing models on an RTX 3090 GPU: UniDepthV2 (94 ms), MoGe (96 ms), DepthAnythingV2 (200 ms); and DepthPro (900 ms). We will include complete data in the final version of the paper.
>
> > Question 1 and 2: Can you describe the rationale for selection of perturbation types and parameters (e.g. occlusion percentage)?
>
> We choose to focus on perturbations that alter the 3D composition of the scene (e.g. object/camera pose, object structure, materials). These typically require re-rendering the original scene and thus are harder to account for via image augmentations at training time. This is in contrast to motion blur or weather effects which are typically implemented as 2D transformations to the image’s pixels.
>
> We choose the parameters of the perturbations so that the depth remains perceptible to a human and the changes to the ground truth are small. For occlusion we observed that 10-40% allowed the object’s shape to remain perceptible but it became difficult at higher percentages. For Camera Dolly Zoom and Object Resizing, we move the camera in order to maintain the 2D size of the object. This preserves the visual information and complexity of the task.
>
> > Question 3: Did you consider alternative metrics (e.g. scale-invariant error) to include affine-invariant models in consistency analysis?
>
> Unfortunately scale-invariant metrics are not sufficient, it must be affine-invariant. Another alternative is to fix a shift and scale, but this can substantially impact the geometry and there is no clear standardization.
> We additionally plan to evaluate self-consistency of affine-invariant models using the affine-invariant relative depth metric between pairs of pixels, which has been adopted by [Yang et al., 2024, Depth Anything V2] and [Chen et al., 2016, Single-Image Depth Perception in the Wild]. Specifically, for model’s prediction of perturbed data d1, and base data, d2, we randomly sample 10^6 pairs of pixels from the object. We report the ratio of pairs such that 1[d1(i)<d1(j)] = 1[d2(i)<d2(j)] (where 1[] is the Iverson bracket notation, (i, j) is one pair of sampled pixels). We will include detailed results in the final version.
>
> > Question 4: AbsRel error may miss local inconsistencies. Did you consider edge aware metrics to evaluate geometric fidelity?
>
> We considered using edge aware metrics (e.g. edge f1 score) but this conflicts with our notion of evaluating the metrics on the object of interest since the edges necessarily include portions of the background. In the appendix (table 3) we include RMSE which also captures local geometric accuracy on the object of interest.
>
> > Question 6: Application to robotics/AR systems
>
> In robotics/AR systems, models can use richer information to improve depth estimation. For example
> Calibrated cameras with known focal lengths can mitigate ambiguity in dolly zoom.
> Using depth sensor readings, stereo input, or videos has also been shown to improve depth estimation [Chen et al., 2024, Video Depth Anything]
>
> > Question 7: Support for custom scenes
>
> We support Blender compatible scenes, but non-rigid object deformation requires procedural assets.
>
> > Question 8: Could this framework evaluate video-based depth estimation?
>
> Yes. Our perturbations can also be applied to generate perturbed videos. Video-based depth estimation metrics (e.g. temporal inconsistency) can be computed and used to calculate the robustness.

---

### Official Review · Reviewer_qXuV · 2025-07-03

**Clarity:** 4
**Significance:** 3
**Originality:** 4
**Rating:** 5
**Confidence:** 4

**Summary:**

This paper introduces Procedural Depth Evaluation (PDE), a new benchmark to assess the robustness of monocular depth estimation models beyond standard accuracy metrics. Using procedural generation, the authors create controlled 3D scene variations (perturbations in object properties, camera parameters, materials, lighting, etc.) to test whether state-of-the-art depth models maintain performance under changes in scene content. Several recent depth models including MiDaS, ZoeDepth, DepthPro, etc. are evaluated on synthetic scene with different types of perturbations, measuring robustness via accuracy stability and self-consistency metrics. The experiments reveal that while models are fairly robust to lighting changes and small 3D pose adjustments, they struggle with certain perturbations– notably occlusions, material swaps, and camera parameter shifts (e.g. camera rotation or focal length changes cause large performance drops). These findings highlight brittle cases for even top-performing depth estimators and underscore the need for improving robustness.

**Questions:**

- Real-World Correlation: Have you evaluated whether the robustness rankings or failure modes observed on synthetic PDE scenes carry over to real-world data? For example, if a model is sensitive to camera roll in PDE, is it also less reliable on real tilted-camera photos? Some confirmation on real images would strengthen confidence in the benchmark's relevance.
- Combined Perturbations: You focused on single perturbations to isolate effects (which makes sense). Did you observe any indications that combining perturbations could be especially problematic? For instance, might a lighting change exacerbate the effect of an occlusion or camera change? It would be interesting to know if any preliminary tests were done or if you have thoughts on how robustness might compound (or not) in some of these cases.
- Camera Perturbation : The finding that ``camera rotation and focal length changes are among the most challenging perturbations'' is interesting. Could you add more discussion to this? Is it due to how depth networks encode perspective (perhaps assumptions baked in from training data) or something like lack of camera calibration handling? DepthPro seems to handle some of these explicitly and how does it help? Understanding the cause might hint at how to make models more robust to these intrinsic/extrinsic changes.
- Improving Robustness: Based on your results, do you have any insights or hypotheses on how to improve model robustness to the difficult perturbations? For example, would data augmentation with certain perturbations, or model architectures that explicitly account for camera intrinsics, help? While it’s beyond the paper’s scope to fully explore solutions, any suggestions for model developers (inspired by your analysis) would be valuable.

**Ethical Concerns:**

["NO or VERY MINOR ethics concerns only"]

**Final Justification:**

The response from the authors is comprehensive and relieved all my previous doubts. The real experiments and discussions on stability variations on SOTA models are very helpful and provide valuable insights. I recommend acceptance of the paper. The authors should organize and include the response in to the main paper should it get accepted.

**Limitations:**

Yes

**Paper Formatting Concerns:**

Figure 1 fails to load in Safari- not sure if it is a browser bug or a PDF compiling issue.

**Quality:**

4

**Strengths And Weaknesses:**

Strengths:
- Novel Benchmark for 3D Scene Perturbations: This work fills an evaluation gap by focusing on scene content changes (object shape/pose, camera intrinsics/extrinsics, lighting, materials) rather than only 2D image corruptions. To the reviewer's knowledge, robustness to such 3D perturbations had not been systematically evaluated before, making PDE a novel contribution.
- Comprehensive and Controlled Evaluation: The benchmark uses 12 distinct perturbation types covering virtually all aspects of scene and imaging variations- from camera zoom/roll/pan, object rotations/translations/resizing, to material swaps, lighting changes, occlusion insertion, non-rigid object deformations, and even out-of-distribution background swaps (e.g. placing a chair underwater). Each perturbation varies only one factor at a time, enabling a clear analysis of its isolated effect on depth predictions.
- Thorough Evaluation of SOTA Models: The authors evaluate 9 state-of-the-art monocular depth models, including both zero-shot and fine-tuned methods, large foundation models (e.g. DepthAnything, UniDepthV2) and highly accurate recent models like DepthPro. The analysis introduces clear metrics (accuracy stability and self-consistency) to quantify robustness. The results are well-documented, with comprehensive tables of error and robustness scores for each model and perturbation, along with qualitative examples. The paper yields insightful findings (e.g. camera rotations and focal length changes degrade performance far more than object pose changes), which are empirically backed by the data, and could guide future methods to improve their designs accordingly.

Weaknesses:
This is a nice and complete work and all these are minor points for dataset improvements:
- Synthetic Domain Only: The evaluation is conducted entirely on procedurally rendered scenes. While the synthetic data is photorealistic and comes with perfect ground truth, there remains a domain gap to real-world images (e.g., textures, sensor noise). The paper justifies this choice and argues humans can interpret the synthetic depths, but it does not demonstrate that models' robustness on PDE correlates with real-world robustness. The lack of any validation on real images or physical scenes means it’s unclear how the reported robustness issues translate to actual deployments. In fact, a small scale capture of such real-world data may be feasible and worth some discussion.
- Limited Scene Diversity: The benchmark focuses on indoor objects and natural outdoor elements, but omits urban driving scenarios- a key domain for depth estimation (e.g. autonomous driving). There are no cars, streets, or building-rich scenes in PDE. Models might exhibit different robustness characteristics in street scenes (e.g. differing scale or motion) which is not evaluated.
- Perturbations Tested in Isolation Only: PDE applies one perturbation at a time to a scene. Combined effects of perturbations (e.g. simultaneous changes in lighting and object pose, or compound weather+camera changes) are not explored. This is a sensible initial simplification to attribute causes. However it remains unknown if interactions between perturbations might further challenge depth models or if the effects simply add up.
- Statistical Analysis and Uncertainty: The paper presents average error and stability metrics over the sampled scenes, but does not report any variability measures (e.g. error bars or statistical significance) for comparisons. Given the results involve many scenes and perturbation instances, it would strengthen the analysis to see variance or confidence intervals on model performance differences, and maybe picking up some more insights. As-is, one must assume the differences (often large) are meaningful, but formal significance tests are not discussed. Incorporating error bars in plots or statistical tests would better support claims that certain perturbations are ``much more difficult'' than others, beyond just avg values.

---

> ### Author Rebuttal · Authors · 2025-07-31
>
> Thanks for your constructive feedback, and we are glad that you found our benchmark novel with “Comprehensive and Controlled Evaluation” and “Thorough Evaluation of SOTA Models”. We have addressed the individual concerns below.
>
> > Concern 1 (and question 1): Results on synthetic data may not generalize to the real world
>
> While we acknowledge concerns about the domain gap, we argue that synthetic evaluation provides invaluable insights that complement measurements with real-world data. Real-world depth faces fundamental limitations that are addressed by synthetic data: (1) challenges obtaining dense ground truth depth in the presence of specular surfaces, transparent objects, and distant objects (2) limited scene variety (e.g. no underwater scenes) (3) prohibitive costs when creating controlled and precise perturbations at a large scale. In the context of our experiments, synthetic data can effectively reveal flaws in models’ performance by enabling precise comparisons for many scenes and perturbations. Notably, this includes perturbations such as object deformation which cannot be easily replicated in the real world.
>
> In addition, our synthetic images are legible to humans in terms of their 3D structure. The fact that models struggle on them reveals potential misalignment of models with human perception, which can help guide model development.
>
> To address this question about generalizability we performed preliminary real-world experiments using a mug with a known 3D structure. We then captured 15-20 images per variation and manually annotated the pose and object mask to infer a precise depth map for the object. We present the results below. Since this is a limited experiment in the context of objects on a table top (similar to the domain of robotic arms), one should not expect the results to perfectly match our much more diverse synthetic evaluation.
>
> **Error**
> | Test Scenario | DepthAnything | DepthAnythingV2 | DepthPro | Marigold | Metric3DV2 | MiDaS | MoGe | UniDepth | ZoeDepth | Average |
> |----|-----|---------|-----|---|---|-------|------|----------|----------|---------|
> | Cam Pan/Tilt | 2.31 | 1.59 | 1.66 | 2.31 | 1.82 | 2.73 | 1.81 | 1.04 | 2.87 | 2.01 |
> | Camera Roll | 1.92 | 1.23 | 1.43 | 2.13 | 1.16 | 1.96 | 1.80 | 1.16 | 2.05 | 1.65 |
> | Lighting | 2.33 | 1.62 | 1.66 | 2.38 | 1.62 | 2.63 | 1.81 | 1.22 | 2.60 | 1.98 |
> | Occlusion | 2.62 | 1.73 | 1.92 | 2.70 | 2.11 | 2.89 | 2.15 | 1.50 | 2.98 | 2.29 |
> | Rotate | 2.28 | 1.73 | 1.76 | 2.75 | 1.91 | 2.65 | 2.15 | 1.52 | 2.76 | 2.17 |
> | Average | 2.29 | 1.58 | 1.69 | 2.45 | 1.72 | 2.57 | 1.94 | 1.29 | 2.65 | 2.02 |
>
> **Stability**
> | Category | DepthAnything | DepthAnythingV2 | DepthPro | Marigold | Metric3DV2 | MiDaS | MoGe | UniDepth | ZoeDepth | Average |
> |----------|---------------|-----------------|----------|----------|------------|-------|------|----------|----------|---------|
> | cam_pan_tilt | 0.1846 | 0.3564 | 0.4183 | 0.3168 | 0.3666 | 0.2016 | 0.3341 | 0.2513 | 0.1217 | 0.2835 |
> | camera_roll | 0.3987 | 0.3414 | 0.3879 | 0.4799 | 0.2666 | 0.3828 | 0.4225 | 0.3122 | 0.5408 | 0.3925 |
> | lighting | 0.2499 | 0.2627 | 0.2516 | 0.4311 | 0.3376 | 0.2002 | 0.3731 | 0.4483 | 0.2947 | 0.3166 |
> | occlusion | 0.4359 | 0.4676 | 0.2591 | 0.4003 | 0.5227 | 0.5714 | 0.3788 | 0.4672 | 0.5548 | 0.4509 |
> | rotate | 0.5684 | 0.4389 | 0.4624 | 0.6950 | 0.3235 | 0.6661 | 0.5379 | 0.3700 | 0.7202 | 0.5314 |
> | avg | 0.3675 | 0.3734 | 0.3559 | 0.4646 | 0.3634 | 0.4044 | 0.4093 | 0.3698 | 0.4464 | 0.3950 |
>
>
> We find similar though not identical trends in terms of models’ average performance, with UniDepth, DepthAnythingV2, and DepthPro remaining as 3 of the top 4 models by average error. We find that occlusion remains a difficult task and many models remain fairly robust to changes in lighting. We will include a more thorough analysis in our final version.
>
> > Concern 2: There is no coverage of urban driving scenarios where models may exhibit different behaviors.
>
> We do not explicitly cover urban/driving scenes, though we do expect our results to generalize to new situations such as these. We evaluate the robustness of the geometry estimation of particular objects, with our selection of objects encompassing different geometric attributes – thin structure (chair, desk), planar shapes (desk, cabinet), irregular curved surfaces (fish, cactus). We expect that a model that can robustly predict these geometries would succeed in an urban driving environment, though we leave this to future work.
>
>
> > Concern 3 (and Question 2): Perturbations are tested in isolation, so the interactions between perturbations are unknown.
>
> We believe testing individual perturbations is sufficient for evaluating robustness. Real-world scenes are not simply “clean” scenes with perturbations added; they naturally contain a mix of challenging factors. Since our base scenes already exhibit substantial diversity, applying perturbations individually already captures a wide range of realistic conditions.
>
> Some of the perturbations (in-distribution perturbations) already exist in the base scenes, e.g., diverse lighting conditions, materials, and camera tilt angles; some of them (out-of-distribution perturbations) do not, e.g., camera roll angles are typically standardized in the base scene. For those in-distribution perturbations, we expect that combining perturbations will have at most a small impact on the average error, while the instability may increase due to the greater difference between the perturbed scene and the original. When combining out-of-distribution perturbations, we would expect the average error to increase, though the exact effect is unclear.
>
> We present preliminary results for combinations of perturbations. Specifically, we generate the variations Camera Pan/Tilt + Translate Object, Camera Roll + Object Material Swap, and Translate Object + Scene Material Swap for chairs and desks. The following tables show the results averaged over all models for the original perturbations and the combined perturbations for the objects chairs and desks.
>
>
> |  | Camera Roll | Object Translation | Object Material Swap | Camera Pan/Tilt | Scene Material Swap |
> |-----------------|-------------|---------------------|------------------------|------------------|----------------------|
> | **Error**       | 2.96 | 2.46 | 2.44 | 2.37 | 2.53 |
> | **Acc. (In)Stability** | 0.57 | 0.36 | 0.35 | 0.41 | 0.47 |
>
> | | Camera Roll + Object Material Swap | Cam Pan/Tilt + Translate Object | Translate Object + Scene Material Swap |
> |---------------|----|---|---|
> | **Error** | 3.20 | 2.46 | 2.58 |
> | **Acc. (In)Stability** | 0.56 | 0.51 | 0.50 |
>
>
>
> This confirms our expectations about the interactions between variations. For variations that introduce out-of-distribution changes, such as camera roll + material swap, the average error compounds when perturbing the scene with both variations. For object translation, which produces in-distribution perturbed scenes, the average error is close to the maximum of the two original errors. We do find for Cam Pan/Tilt + Translate Object that the instability increases, which is sensible since the difference from the original image is now larger.
>
> > Concern 4: The paper does not report error bars or statistical significance for comparisons. Incorporating error bars in plots or statistical tests would better support claims that certain perturbations are ``much more difficult'' than others, beyond just avg values.
>
> In this case, it is unclear exactly how we could report meaningful error bars. We could attempt to understand the randomness in the models’ scores with respect to the data generation process by generating different versions of our dataset. However, we believe this to be less useful than spending time generating and verifying the quality of a larger dataset. Alternatively, could compute error bars for the average perturbation difficulty (error or stability) while treating the model outputs as samples. However, these error bars would neglect the relations between the model predictions for different perturbations. Since we currently have only 9 models we don’t believe these error bars would provide much insight.
>
> > Question 3: Why are camera perturbations particularly challenging and why are some models more robust to camera changes?
>
> It is somewhat surprising that camera perturbations are challenging, and the success of UniDepthV2 and DepthPro suggests that an explicitly camera model is important to achieve robustness. For camera dolly zoom, we see a large difference in stability between the top models (Depth Pro and UniDepthV2) compared to the average of all models, while such phenomenon is not seen for other perturbation types. Depth Pro and UniDepthV2 both explicitly handle camera models. DepthPro predicts canonical inverse depth, and UniDepthV2 disentangles camera and depth prediction. Notably UniDepthV2's predictions of camera roll are much more stable than DepthPro. This might suggest that predicting dense camera representation is better than sparse representation (focal length).
>
> > Question 4: What are some insights or hypotheses or improving model robustness to the difficult perturbations? Would data augmentation or certain model architectures help?
>
> We suspect that explicit augmentation would improve performance for Camera Roll, as such data is fairly rare. For other challenging variations (e.g. material changes or occlusion) we expect it would be more valuable to utilize challenging high quality data sources which may be real or synthetically generated. Another possible hypothesis is that a local geometry loss similar to MoGe would improve model’s robustness, as MoGe performs exceptionally well on the self-consistency score. However, its average error still falls behind that of Depth Pro.
>
> > Formatting: Figure 1 fails to load in Safari
>
> Thanks for pointing this out. We have adjusted figure 1 and verified that it works on all major desktop and mobile browsers.

---

> > ### Comment · Reviewer_qXuV · 2025-08-08
> >
> > Thanks for the response. I find the new experiments and discussions very helpful and is convinced this paper would provide valuable insights into the design of future depth models.

---

### Decision · Program_Chairs · 2025-09-17

**Decision:**

Accept (poster)

**Comment:**

The final ratings are 3 borderline accepts, 1 accept. The AC have read the reviews and rebuttal, and discussed the submission with the reviewers. The reviewers raised a number of points during the review phase including limitations to synthetic data, limited scene diversity and rational/analysis of the experiments. The authors were able to address these points during the rebuttal and discussion and swayed a previous borderline reject to borderline accept. The AC recommends the authors to incorporate the feedback and suggestions provided by the reviewers, and the materials presented in the rebuttal, which would improve the next revision of the manuscript.